# Stream hydrology controls on ice cliff evolution and survival on debris-covered glaciers

Eric Petersen[1], Regine Hock[2,1], and Michael G. Loso[3]

[1]Geophysical Institute, University of Alaska Fairbanks, Fairbanks, AK, USA
[2]Department of Geosciences, University of Oslo, Oslo, Norway
[3]Wrangell-St. Elias National Park and Preserve, National Park Service, Copper Center, AK, USA

**Correspondence:** Eric Petersen (eipetersen@alaska.edu)

**Abstract.** Ice cliffs are melt hot spots that contribute disproportionately to melt on debris-covered glaciers. In this study, we investigate the impact of supraglacial stream hydrology on ice cliffs using in-situ and remote sensing observations, stream flow measurements, and a conceptual geomorphic model of ice cliff backwasting applied to ice cliffs on Kennicott Glacier, Alaska. We found that 33% of ice cliffs (accounting for 69% of ice cliff area) are actively influenced by streams, while half are less than 10 m from the nearest stream. Supraglacial streams contribute to ice cliff formation and maintenance by horizontal meandering, vertical incision, and debris transport. These processes produce an undercut lip at the ice cliff base and transport clasts up to tens of centimeters in diameter, preventing reburial of ice cliffs by debris. Stream meander morphology reminiscent of sedimentary river channel meanders and oxbow lakes produces sinuous and crescent ice cliff shapes. Stream avulsions result in rapid ice cliff collapse and local channel abandonment. Ice cliffs abandoned by streams are observed to be reburied by supraglacial debris, indicating a strong role played by streams in ice cliff persistence. We also report on a localized surge-like event at the glacier's western margin, which drove the formation of ice cliffs from crevassing; these cliffs occur in sets with parallel linear morphologies contrasting with the crescent planform shape of stream-driven cliffs. The development of landscape evolution models may assist in quantifying the total net effect of these processes on steady state ice cliff coverage and mass balance, contextualizing them with other drivers including supraglacial ponds, differential melt, ice dynamics, and collapse of englacial voids.

## 1 Introduction

Glaciers covered extensively by rocky debris are found worldwide but are most common in High Mountain Asia, Alaska, and the Central Andes where glaciers are key contributors to glacial melt and regional freshwater resources (Scherler et al., 2018; Herreid and Pellicciotti, 2018; Bhushan et al., 2018). It is thus critical for these regions to have a firm understanding of the effect supraglacial debris has on surface melt.

Supraglacial debris exhibits a strong control on glacier melt which is empirically described by the Østrem curve (Østrem, 1959). Thin debris cover increases melt relative to a bare ice surface, but a sufficiently thick debris layer (typically on the order of a few centimeters for rocky debris) attenuates melt significantly in a non-linear fashion (Østrem, 1959; Khodakov, 1972; Dolgushin, 1972; Bozhinskiy et al., 1986; Mattson et al., 1993). Debris-covered glaciers are defined by the development of

continuous debris cover across much of their ablation zone (Cogley et al., 2011), with debris commonly thick enough to retard ablation and lead to the development of a stagnant tongue of ice with a delayed response to climatic forcing (Benn et al., 2012; Bolch et al., 2012; Mölg et al., 2020; Anderson and Anderson, 2016).

Ice exposed at the surface of ice cliffs contributes significantly to melt on debris covered glacier surfaces (Sakai et al., 2002; Buri et al., 2021; Anderson et al., 2021a). Miles et al. (2022) determined that melt rates on ice cliffs are consistently 2-3× melt rates on clean glacier ice under similar conditions. As a result, ice cliffs which cover ∼10% of the area of debris-covered glacier tongues contribute ∼20-25% of the melt rates over that same area, as shown for glaciers in Alaska (Anderson et al. (2021a): 12% coverage, 26% melt rates on Kennicott Glacier) and Nepal (Brun et al. (2018), 7-8% area coverage, 24±5% melt rates on Changri Nup Glacier). Ice cliff melt is also significant on glacier and catchment-wide scales; Buri et al. (2021) found in a modeling study that ice cliffs account for 17±4% of total glacier melt across the Langtang Glacier catchment.

Some studies have thus suggested that ice cliffs contribute to the "debris-cover anomaly" which describes the observation that area-averaged melt rates are comparable for debris-covered and non-debris covered portions of glaciers at similar elevations, despite the localized attenuation of melt by thick debris cover (Kääb et al., 2012; Gardelle et al., 2013; Pellicciotti et al., 2015; Maurer et al., 2019). In a study of two Tibetan debris covered glaciers Zhao et al. (2023) determined that although ice cliffs enhanced melt overall (with a melt enhancement factor of ∼2.5), they did not play a role in controlling surface mass balance patterns or patterns of thinning. Rowan et al. (2021) determined that melt hot spots including ice cliffs can increase mass loss on Himalayan glaciers by 29-47%, however this is still not sufficient to explain the debris cover anomaly, which was attributed by the authors to ice dynamics. Regardless, it is of importance to understand and capture ice cliffs in debris-covered glacier melt models.

Controls on ice cliff formation, evolution, and survival are therefore an area of intensifying research (Kneib et al., 2023; Sato et al., 2021; Buri and Pellicciotti, 2018; Anderson et al., 2021a; Watson et al., 2017b, a). Differential melt under variable debris thickness creates rough surface topography (Anderson, 2000; Moore, 2021; Westoby et al., 2020; Bartlett et al., 2020); ice cliffs may be generated where the surface is sufficiently steep to slough debris. Ice cliffs may also be produced as a result of modified crevasses, from the incision of supraglacial streams, from the collapse of englacial voids resulting in exposed ice walls (Benn et al., 2001; Mölg et al., 2020), or be associated with the rims of supraglacial ponds which may prolong their lifespan (Watson et al., 2017a). Buri and Pellicciotti (2018) showed that aspect is a strong control on cliff survival; ice cliffs with a poleward aspect tend to persist whereas ice cliffs with an equatorward aspect tend to flatten as a result of differential insolation-driven melt, and are reburied.

A number of studies have suggested a strong relationship between ice cliffs and supraglacial streams. Mölg et al. (2020) noted on Zmuttgletscher the presence of "cryo-valleys" carved into the debris-covered surface by supraglacial streams, with ice cliffs formed in the resulting high relief. Anderson et al. (2021b) noted that ice cliffs often act as initiation points for supraglacial streams due to their role as a locally enhanced melt water source. Sato et al. (2021) reported a correlation between active supraglacial stream networks and the presence of new ice cliffs, suggesting a causative link. Kneib et al. (2023) noted that 38.9% of ice cliffs surveyed on 86 glaciers in High Mountain Asia were associated with supraglacial streams, while 19.5%

were associated with supraglacial ponds. This highlights a close relationship between ice cliffs and hydrology, particularly
stream hydrology.

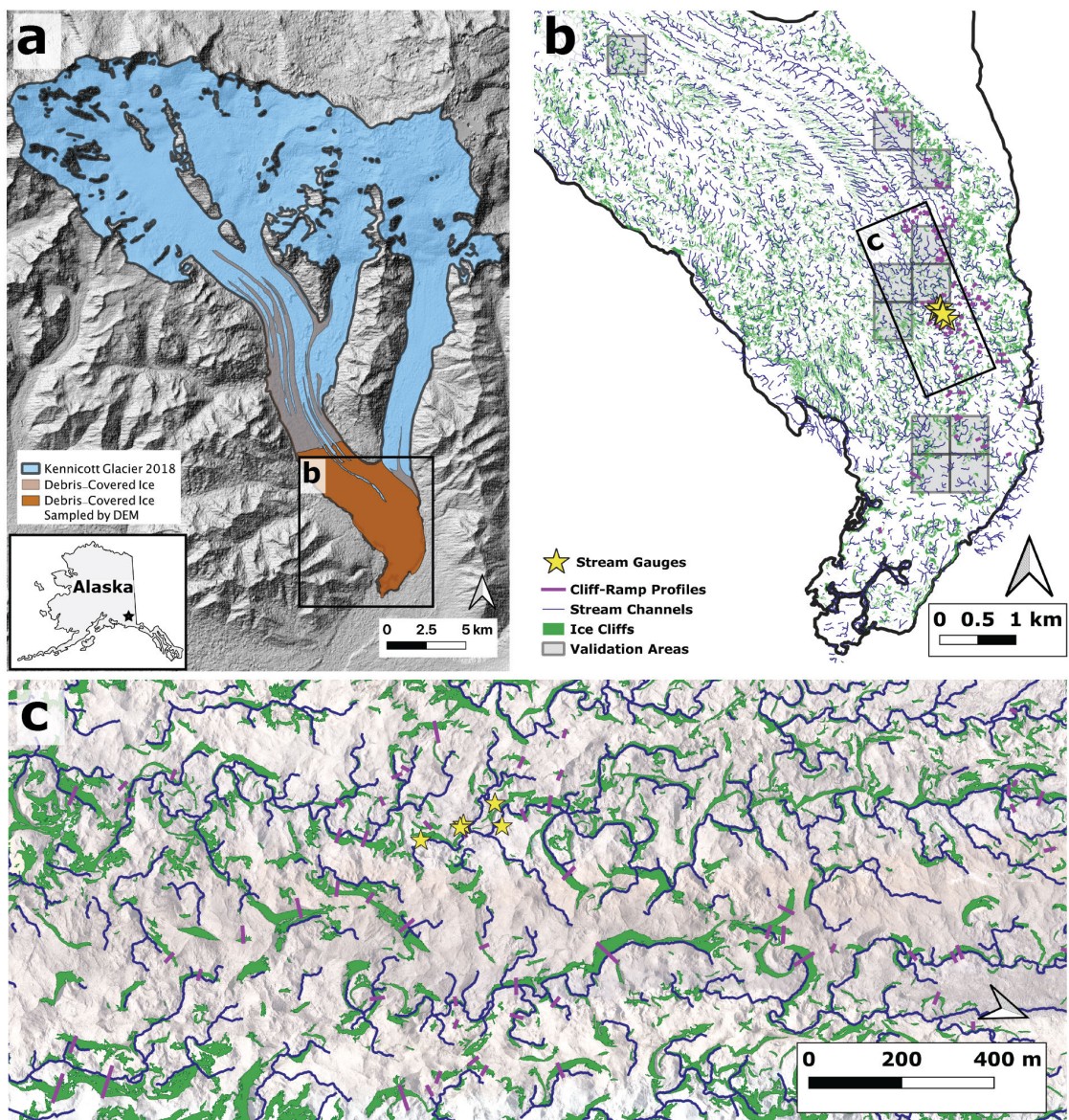

**Figure 1.** (a) Kennicott Glacier and tributaries with outlines mapped on an ASTER (Advanced Spaceborne Thermal Emission and Reflection Radiometer) DEM hillshade (Spacesystems and Team, 2019). Debris-covered area as well as the area sampled by the airborne photogrammetry DEM presented in this study is mapped. (b-c) Delineated ice cliffs and stream channels (both predicted from DEM analysis) as well as the location of discharge measurements. Validation areas mapped in (b) refer to areas where automatically mapped ice cliffs and stream channels were compared to manually mapped results. Panel (c) includes airborne photogrammetry orthophoto as a semi-transparent basemap (National Parks Service Data, presented in this study). Arrows in each panel represent north direction.

In this study, we examined in detail the links between ice cliffs and supraglacial stream hydrology on Kennicott Glacier, Alaska. We performed analysis on a photogrammetric dataset (September 2018) and in-situ observations (summer field seasons 2020-2022), and developed a conceptual geomorphic model of ice cliff backwasting to examine stream impacts on ice cliff development. We also describe the results of stream gauging to determine the ability of streams to erode and transport supraglacial debris.

## 2   Study site

Kennicott Glacier is a 43 km long valley glacier located in Wrangell-St. Elias National Park, Alaska (Figure 1). It flows from the southern flank of Mount Blackburn at 4996 m a.s.l. to its terminus at $\sim$400 m a.s.l., and is centered on 61.573° N, -143.043° E (RGI Consortium, 2023). Instantaneous horizontal surface velocities on the mid section of the glacier have been measured by GPS from 0.14-0.95 m d$^{-1}$ (Bartholomaus et al., 2011), and yearly average surface velocities measured by ITS_LIVE in the lower half of the glacier range from $\sim$220 m yr$^{-1}$ to fully stagnant in the lowermost 2.5-3 km of the glacier (Gardner et al., 2022, 2018). Nine medial moraines grow and merge down glacier, causing the formation of a continuous supraglacial debris layer $\sim$7 km from the terminus, with a debris cover typically tens of cm (but <50 cm) thick and subdebris melt rates of 1-7 cm d$^{-1}$ (Anderson et al., 2021a). Anderson et al. (2021a) estimated that 20% of Kennicott Glacier's surface is debris-covered; on the lowermost 8.5 km by length of Kennicott Glacier they determined that 26% of melt was attributed to ice cliffs covering 12% of the surface in summer 2011. The same study mapped supraglacial ponds and found that they were most numerous and covered the most area in the stagnant, lowermost $\sim$2-3 km of the glacier, outside of which they were only abundant on medial moraines near the glacier margins where debris thickness was thicker and surface velocities lower.

## 3   Methods

### 3.1   Ice cliff mapping using photogrammetry

The Wrangell-St. Elias National Parks Service (NPS) commissioned a photogrammetric dataset, flown on September 13, 2018 over the debris-covered terminus of Kennicott Glacier from its confluence with the Root Glacier to the toe. 1691 images were taken using a Nikon D850 with a 24mm prime lens at an average flying altitude of 940 m above sea level. These images were processed using Agisoft Metashape to produce a digital elevation model (DEM) at 0.32 m resolution and an orthophoto at 0.16 m resolution. Four ground control points were surveyed at the McCarthy airport (61.4377° N, 142.9027° W, $\sim$1.5 km distance from the glacier terminus) to calibrate the final DEM product, resulting in a +0.13 m vertical adjustment to the ellipsoidal heights. The photogrammetric dataset covers 33.8 km$^2$ (37.2 km$^2$ when corrected for slope) of the total 54.8 km$^2$ of debris-covered ice on Kennicott Glacier, and extends up to an elevation of 830 m above sea level on the glacier surface (coverage mapped in Figure 1a).

Past efforts at ice cliff mapping have included manual mapping using visible imagery (Brun et al., 2018; Zhao et al., 2023) at times supplemented by slope maps (Steiner et al., 2019), and automated mapping using spectral curvature (Kneib et al.,

2020), object/image segmentation algorithms applied to visible imagery (Kraaijenbrink et al., 2016; Anderson et al., 2021a), and slope thresholds using a DEM as input (Herreid and Pellicciotti, 2018). While manual mapping is expected to provide the highest fidelity data product, automatic mapping is more feasible for large areas such as at Kennicott Glacier. Automated mapping via visible/spectral image analysis has the advantage that it can be applied to a wide range of orbital/remote sensing datasets. Automated mapping via slope threshold has the advantage of relative simplicity and is not sensitive to variable lighting conditions or albedo of ice cliffs; further it is based on the foundational principle of ice cliffs' existence — slopes greater than the angle of repose have no debris layer. We developed our own automated mapping algorithm based on the slope threshold method.

Our automated ice cliff mapping algorithm uses slope calculated from the photogrammetric DEM as follows. (1) Grid cells with slope angles greater than $> 30°$ (representing areas where the angle of repose is exceeded) were isolated in a mask. (2) This mask was converted to a shapefile with contiguous collections of grid cells as individual shapes. (3) Gaps $< 10$ m$^2$ in size contained entirely within ice cliff shapes were filled. (4) We then calculated the steepest slope value contained within each individual ice cliff shape; a threshold minimum steepest slope value was tested as a method of reducing false positive identification of ice cliffs.

As a validation for our automated ice cliff mapping method, we performed manual mapping of ice cliffs via visual identification on a subset of the orthophoto. The debris-covered terminus region of the glacier was sliced into 500 m x 500 m grid cells and 11 of these were selected for the mapping validation, sampling 8% of the total orthophoto (Figure 1b). Automatically identified cliffs that coincide with manually mapped cliffs were marked as "true positive," while those which did not were marked as "false positive." We also calculated the F1 score (Dice, 1945; Manning, 2009), which combines metrics of false positive and false negative identification:

$$F1 = \frac{2TP}{2TP + FP + FN} \tag{1}$$

where TP, FP, and FN are the true positive, false positive, and false negative fractions, respectively. Our goal is to balance false positive with false negative identifications, while maximizing true positives.

## 3.2 Stream mapping using photogrammetry

To prepare the DEM for hydrologic analysis, it was downsampled to 2 m resolution in order to reduce computation time. Supraglacial streams often undercut ice cliffs and are thus not visible in the top-down view of the photogrammetry dataset; debris mounds which the stream channel meanders around (via ice cliff undercut) may be seen as hydrologic obstructions. For this reason sinks $< 3$ m in depth were filled in the DEM; this value was chosen as a cutoff based upon the typical roughness of debris-covered ice within stream corridors observed in the field on Kennicott Glacier. Stream channels were then predicted over the modified DEM by using the Channel Network tool in SAGA-GIS; this tool calculates runoff catchment area for each pixel in the DEM and returns channels defined by contiguous pixels with catchment area greater than a set threshold, defined in this case as 2500 m$^2$. The chosen threshold was somewhat arbitrary, but represents a potential channel flow of $\sim$1200-2300 cm$^3$

$s^{-1}$ (assuming a mean sub-debris melt rate of 4–8 cm per day and ignoring ice cliffs). Note that these are predicted channels, and may not necessarily be filled with melt water in all cases.

The validation grid cells used in ice cliff mapping (Section 3.1) were also used for a validation exercise of automated stream channel mapping. Each stream channel delineated by the flow algorithm was examined in the orthophoto and categorized according to the following scheme: 1 – channelized/flowing water identified, 2 – channel morphology identified; no water, 3 – no clear channel identified, but plausible (textured debris, association with hydrological features such as ponds and channels, part of dendritic network), 4 – implausible/no hydrologic morphology identified, 5 – error (e.g. channel within a supraglacial pond) (Examples in Supplementary Figure S1).

We then determined the spatial proximity between supraglacial streams and ice cliffs at 2 m resolution. For each grid cell representing a 2 m length of supraglacial stream, the distance to the nearest ice cliff grid cell was calculated, and vice versa. Distances less than 2.83 m represent adjacent grid cells and thus direct ice cliff-stream interactions.

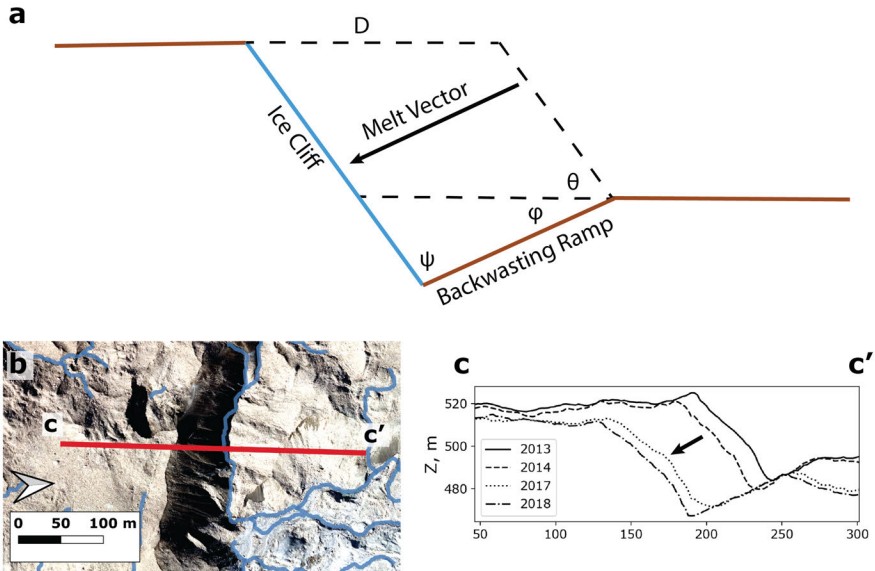

**Figure 2.** (a) Conceptual model for the geometry of ice cliff backwasting. The melt vector is orthogonal to the exposed ice surface (angle $\theta$ from the horizontal), thus there is a component of vertical incision in addition to horizontal backwasting. The ice cliff as it backwastes produces a ramp of angle $\psi$ to itself and $\phi$ to the horizontal. The value of the angle $\psi$ indicates the efficiency of incision. (b) Map illustrating the location of profile c-c' over a large ice cliff associated with a supraglacial stream (mapped as blue lines). (c) Topographic profiles extracted from ArcticDEM illustrating the backwasting of the ice cliff and the carving out of the backwasting ramp. This ice cliff has an angle between cliff and ramp of $\psi = 110°$. This ice cliff is in the glacier's stagnant terminus region where the surface velocity is negligible (values of <10 m $yr^{-1}$ comparable to values determined outside the glacier outline) and thus did not need to be deconvolved from the signal of surface change.

## 3.3 Geomorphic model of ice cliff backwasting

We developed an idealized geomorphic model of ice cliff backwasting in order to analyze the effect of streams on ice cliffs as recorded in the DEM. This model focuses on the relationship between ice cliffs and their "backwasting ramps." We define a backwasting ramp as the slope on a debris-covered glacier surface generated by the ice cliff melting vertically as well as horizontally into the surface of the glacier. While backwasting ramps have not previously been identified in the literature, this model is similar in concept to that presented by Evatt et al. (2017) for ice sails (positive relief bare ice features on some debris-covered glaciers).

Consider an ice cliff at an angle of $\theta$ that backwastes through a two-dimensional debris-covered glacier surface (Figure 2a). Backwasting is driven by melt orthogonal to the ice cliff face. Thus, as the ice cliff melts at a rate $M$ it produces vertical incision ($M cos(\theta)$) as well as horizontal wasting ($M sin(\theta)$), and scours out a backwasting ramp at an angle of $\pi/2$ from the ice cliff and $\phi = \pi/2 - \theta$ from the horizontal. However, incision at the base of the ice cliff is not expected to be fully efficient, due to the effects of differential insolation-driven ablation and reburial by debris. The presence of streams may also aid in the efficiency of incision. We define the angle $\psi$ between the ice cliff and its backwasting ramp. $\psi$, as well as the ramp angle $\phi$, are theoretically indicative of the efficiency of ice cliff incision and survival.

We first sought validation for our model of ice cliff backwasting and ramp scouring by investigating a time series of ArcticDEM data strips (Porter et al., 2022). The ArcticDEM is constructed from stereo pairs of WorldView and GeoEye satellite images, processed using "Surface Extraction by TIN-based Search space Minimization" to DEMs with 2 m horizontal resolution. For the years 2013 (July 15), 2014 (July 21), and 2017 (August 9), available DEM products covering Kennicott Glacier during the summer months of June-August were extracted using Google Earth Engine. These were then coregistered by minimizing the mean elevation difference between three patches of flat, stable, lightly vegetated glaciofluvial or moraine deposits in each DEM mosaic, applying a simple elevation offset to do so. From our investigation of stable terrain we estimate the uncertainty due to horizontal shift between the DEMs at 2-3 m (on the order of the horizontal resolution).

We then extracted topographic profiles for analysis of 100 ice cliffs and their associated backwasting ramps, using the 2018 high resolution photogrammery DEM (profile locations shown in Figure 1b-c). We sought to investigate differences in cliff-ramp geometry between stream-associated cliffs and non stream-associated cliffs, in the context of our geomorphic model. Ice cliffs were therefore categorized as those associated with streams and those with no associated streams. The stream channel prediction results were used to aid in this categorization but the presence of active stream action was confirmed in each case via visual identification of water flow or ice cliff undercutting visible in the orthophoto. Great care was taken in sampling the backwasting ramps, such that our sampling was not affecting by other disturbances such as surface collapses, debris movement, or other ice cliffs. The average slope of the ice cliff and ramp were then determined, along with the corresponding value of $\psi$ between them.

To assess the impact of topographic profile selection on the results obtained using our geomorphic model, we selected two of the sampled ice cliffs for which we extracted 12-16 topographic profiles along their length. Each profile was independently

analyzed using our model, and by comparing the results we determined an uncertainty due to profile selection of 3° for ice cliff and backwasting ramp angles and 5° for $\psi$ (Supplementary Text S1, Figure S2).

### 3.4 Stream discharge measurements

We performed a series of discharge measurements to determine the ability of supraglacial streams to erode and transport debris. We selected a stream network exhibiting numerous examples of cliff-stream interactions for these measurements. We used the area-velocity method (Herschy, 1993) and nine measurements were performed at five locations (locations shown in Figure 1). Experiments took place during afternoon hours across three days in 2021 representing hot, sunny (July 13, 17) and cool, cloudy (July 14) weather conditions. Note that because of the timing of the measurements the results are most indicative of diurnal maxima of flow conditions. Stream measurement locations were selected in reaches with linear, laminar flow and minimal turbulence. The length of sampled reaches varied between 1.5 and 8 m.

Two to three flow-transverse gates were defined for each discharge measurement and the channel cross section for each gate measured by hand using a measuring stick at 10 cm intervals. The area for each discharge measurement was then taken as the average of the area for the measured gates.

A neutrally buoyant mandarin orange was timed as it travelled in the stream between the flow-transverse gates. Individual measurements were removed from the analysis when the orange experienced undue turbulence or friction by interacting with eddies, the channel sidewall, or obstacles such as rocks in the stream bed. For each experiment 5-15 velocity measurements were made and averaged. Full information on individual reaches is summarized in Supplementary Table S1.

Average velocity across the gates is estimated at $0.9\times$ the value of the measured velocity. This is consistent with low friction stream bed such as an artificial channel or in this case bare ice (Rantz, 1982). The determined average velocity is then multiplied by the measured gate area to estimate discharge.

## 4 Results

### 4.1 Ice cliff mapping using photogrammetry

Without applying a minimum steepest slope criteria, 78% of ice cliffs were identified as true positive, while 22% were false positive (steep sections of debris covered ice). Applying a minimum steepest slope criteria reduces the number of false positives, while the false negative fraction begins to increase as the minimum steepest slope threshold is increased (Figure 3f). A threshold value of 51.8° yields an area-averaged balance between false positives and false negatives at 4% each. The maximum true positive fraction and F1 scores occur at 51.7° with values of 93% and 0.96 respectively. This threshold value is also consistent with the distribution of values from manually mapped ice cliffs; the maximum slope observed on 90% of ice cliff shapes is >50° (Figure 3e). We selected a final value of 51.8° for the minimum steepest slope threshold in our automated ice cliff mapping routine.

This threshold for the minimum steepest slope value in each candidate ice cliff shape reduces the number of false identifications from steep debris-covered slopes or fully reburied ice cliffs (Figure 3). Some shallower cliffs are lost as a result but these

are a comparatively small fraction of the population. Because partially reburied ice cliffs are liberally mapped by the automated slope method, the total area compared to the manually mapped results is overestimated by 13%.

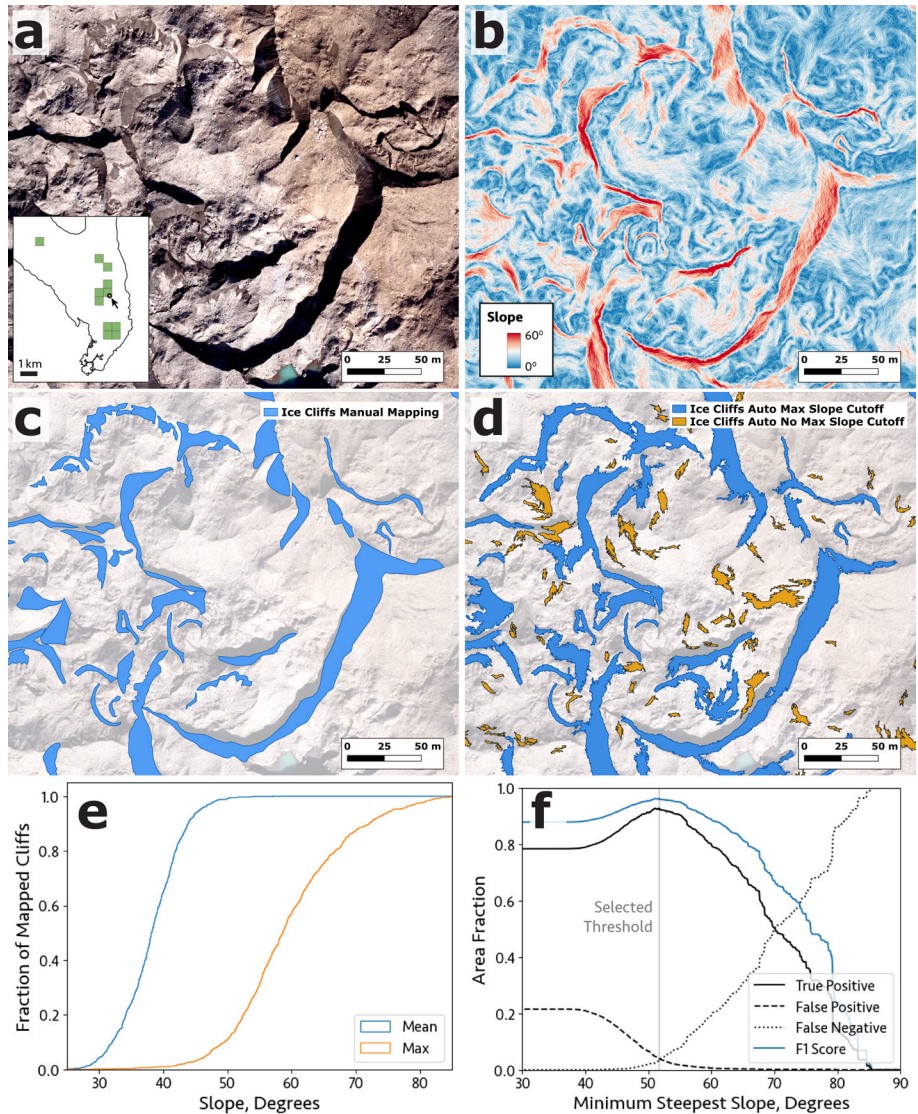

**Figure 3.** Ice cliff mapping validation using airborne photogrammetry. (a) Orthophoto sample of debris-covered Kennicott Glacier surface featuring ice cliffs. Inset: location of sampled area on lower Kennicott Glacier indicated by black arrow and circle (validation areas mapped in green). (b) Slope map derived from Digital Elevation Model. (c) Results of manual ice cliff mapping based off orthophoto. (d) Results of automatic ice cliff mapping as a result of masking out all pixels with slopes greater than $30°$. The effect of including a required minimum steepest slope threshold ($51.8°$) for contiguous ice cliff shapes is shown. (e) Cumulative distribution of mean and maximum slopes on manually mapped ice cliff shapes. (f) Fraction of true positive and false positive identifications of ice cliffs using the automated slope method, as a function of minimum steepest slope threshold for individual ice cliffs. The selected threshold ($51.8°$) is taken where the false positive fraction is equal to the false negative fraction, and also is near the maximum true positive fraction and F1 scores (93% and 0.96 respectively, at $51.7°$).

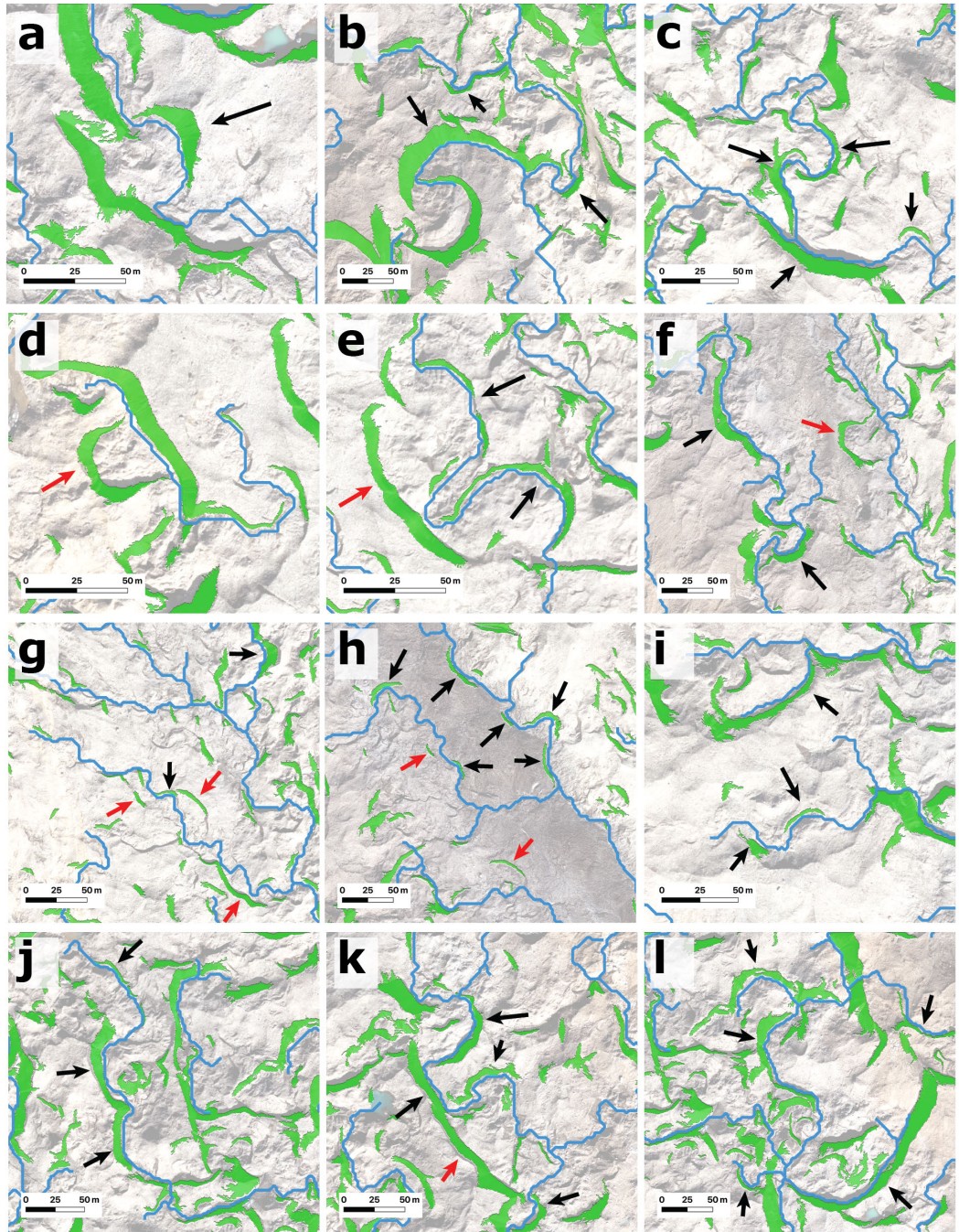

**Figure 4.** Examples of spatial relationships between identified ice cliffs (green) and stream channels (blue) mapped over a low-opacity orthophoto (photogrammetry dataset, Section 3.1). Black arrows indicate curving ice cliff morphologies observed on the outside bend of supraglacial stream meanders; red arrows indicate similar ice cliff morphologies which appear to have been abandoned by stream channels.

The automated ice cliff detection routine identified 10,306 ice cliffs with a total combined area, corrected for slope, of 5.22 $km^2$ (planform area 3.95 $km^2$). Ice cliffs thus account for 14.0% of the 37.2 $km^2$ sampled debris-covered surface (11.7% when ignoring slope in the area calculation). Median ice cliff surface area (true area) is 105 $m^2$ (IQR 45-290 $m^2$). Median surface slope on individual ice cliffs ranged from 32-54° (Supplementary Figure S3a). Cliffs were observed at all aspects, with broad peaks in the population facing NE and SW (Supplementary Figure S3c).

## 4.2 Stream mapping using photogrammetry

A total of 5892 supraglacial stream channels were delineated, with lengths ranging from 4-1241 m and a total cumulative length of 51.9 km (average stream density 1.4 km $km^{-2}$ for the study area). Average stream length was 88 m, with a standard deviation of 79 m, and median of 65 m (Supplementary Figure S3d).

30% of delineated stream channels were associated with identified active stream flow, 44% were associated with channel morphology, 19% were identified as plausible but without clear channel morphology, 5% were identified as implausible, and 1% were identified as artifacts that occurred within large supraglacial ponds (see examples in Supplementary Figure S1). Note that channels without observed active stream flow may be recently abandoned, or they may simply exhibit flow which is below the detection limit dictated by the resolution of the orthophoto (several pixels; ∼1 m), or is frequently in shade/hidden by ice cliff undercuts. Thus, it is not possible to accurately estimate the proportion of active vs. inactive/abandoned stream channels from these results.

These validation results indicate 74% true positive identification of supraglacial streams. With an additional 19% plausible identifications, we determine a possible 6-25% overestimation of stream length/prevalence across the glacier surface.

Stream channel locations frequently coincide with the base of ice cliffs (Figures 4-5). 33% of ice cliffs are directly adjacent to streams while 50% are within 10 m of streams (Figure 5). Cliffs adjacent to supraglacial streams tend to be larger than those at a distance from streams; 69% of ice cliff area belongs to ice cliffs adjacent to streams. Conversely, 20% of total stream length (evaluated in 2 m grid cells/stream lengths) is directly adjacent to ice cliffs, while 48% of stream length is within 10 m distance from the nearest ice cliff. Stream channel networks connect together numerous ice cliffs with sub-linear, sinuous, and crescent planform morphologies. Stream course meanders are frequently associated with distinctive crescent-shaped ice cliffs on the outside of stream bends, reminiscent of cut banks on river channel systems in sedimentary environments. Examples of such ice cliffs are labelled in each panel of Figure 4 with black arrows. Crescent ice cliffs are also frequently observed at a distance of some tens of meters from delineated stream channels (labelled in each panel of Figure 4 with red arrows), implying a historical connection to the stream channel system.

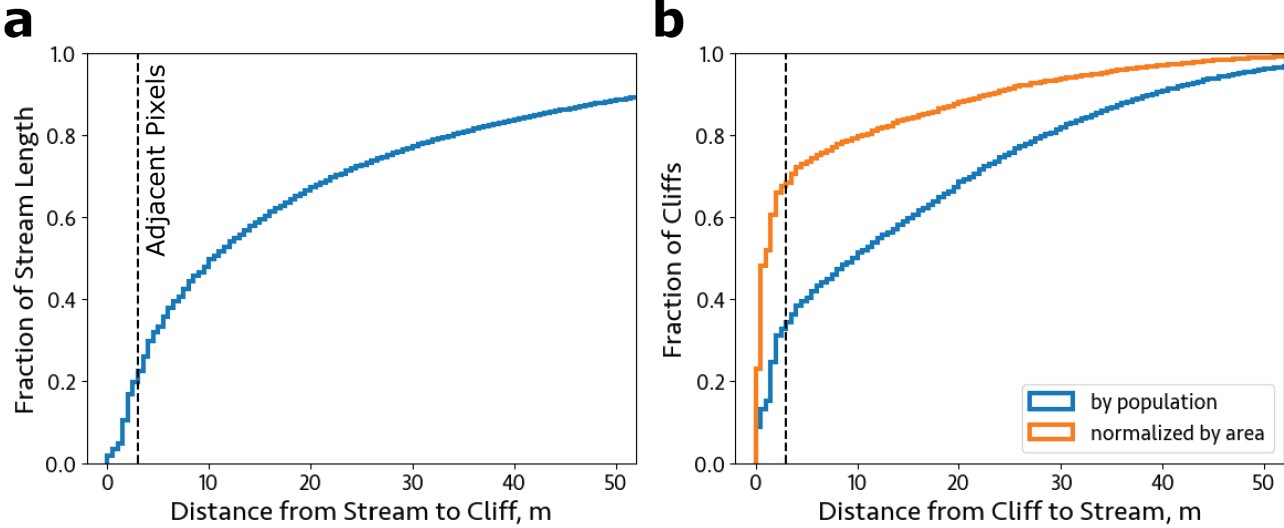

**Figure 5.** (a) Cumulative fraction of supraglacial stream length found within a given distance of identified ice cliffs. Threshold distance for adjacent DEM pixels is plotted. 30% of total stream length is directly adjacent to ice cliffs, while 51% is <10 m from ice cliffs. (b) Cumulative fraction of ice cliffs found within a certain distance of supraglacial streams; 33% of cliffs (accounting for 69% of ice cliff area) are directly adjacent to streams while 46% (80% of ice cliff area) are within 10 m of streams.

### 4.3 Geomorphic model of ice cliff backwasting

We selected a very large (∼40 m tall) ice cliff in the stagnant ice zone of Kennicott Glacier (lower most ∼2.5 km where glacier velocities measured in ITS_LIVE fall to values similar to noise floor observed outside glacier outlines (Gardner et al., 2022)) to investigate ice cliff backwasting through time. This minimized the relative impact of surface change due to glacier flow. As shown in Figure 2b-c, the ice cliff can be seen backwasting and scouring out a ramp, in this case at an angle of $\psi$ = 110 °, consistent with our geomorphic model. This ice cliff is unique among the large ice cliffs in Kennicott Glaciers' stagnant zone in that it has a supraglacial stream and debris covered ramp at its base, and not the surface of a supraglacial pond.

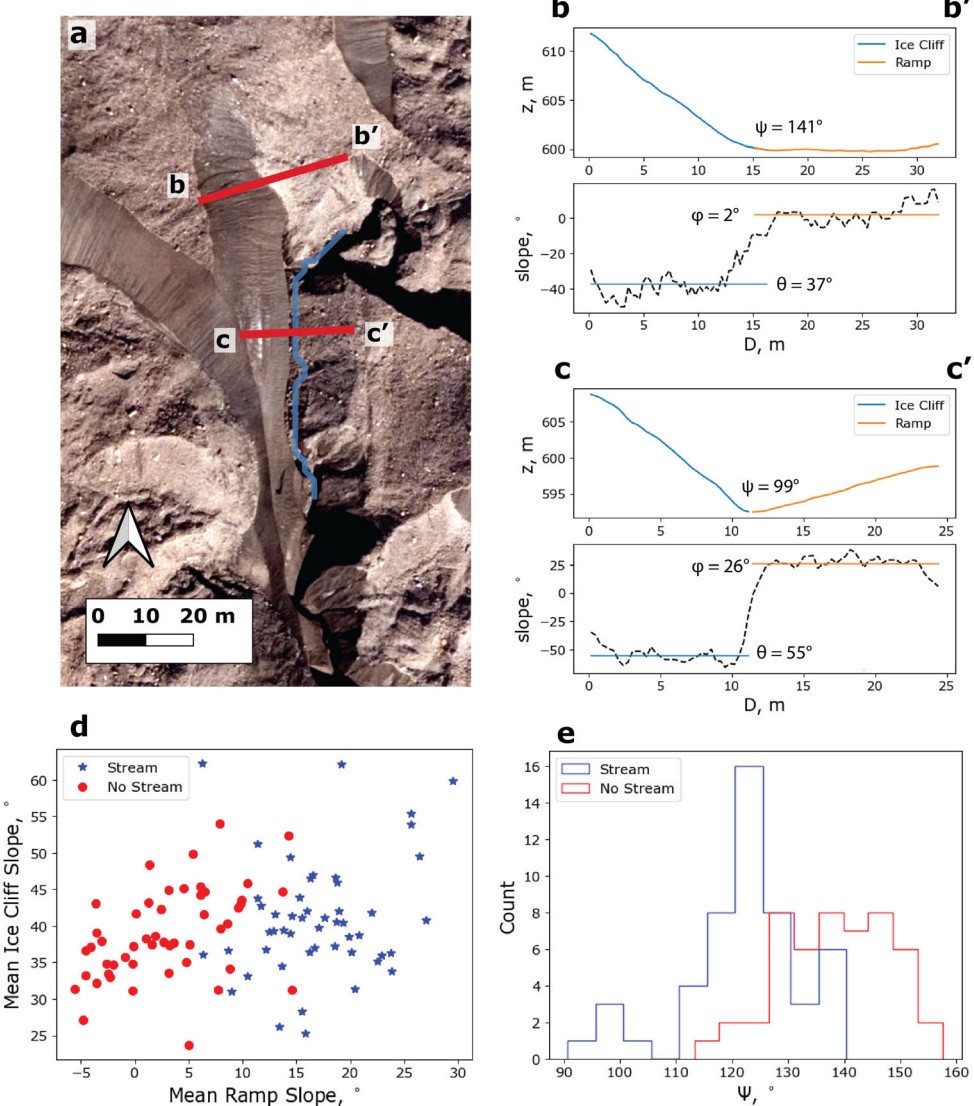

**Figure 6.** Ice cliff backwasting ramp analysis. (a) Orthoimage illustrating the location of profiles b-b' and c-c' on an ice cliff which experiences partial undercutting by a supraglacial stream. (b) Topographic profile over the ice cliff and backwasting ramp where it is unaffected by the supraglacial stream, resulting in $\psi = 141°$. (c) Topographic profile where the ice cliff is undercut by the supraglacial stream, resulting in $\psi = 99°$. (d) Ice cliff slope $\theta$ plotted against ramp slope $\phi$ for 100 profiles of systems with and without associated streams. (e) Histograms of $\psi$ values for ice cliffs with and without associated streams.

We sampled 100 cliff-ramp systems with roughly equal shares of cliffs with and without streams to investigate the impact of streams on the backwasting. Two example ice cliff - ramp profiles are shown in Figure 6a, taken from the same ice cliff. Profile b-b' is taken from a location where the cliff has no associated stream at its base; to the north of the profile the cliff appears to

be in the process of reburial under debris. Profile c-c' is taken from a location where a supraglacial stream channel is predicted, and corroborated by geomorphic evidence in the presence of an incisional canyon that approaches and intersects the ice cliff from the east, as well as the sharp, likely undercut base of the cliff. Profile c-c' exhibits a sharper $\psi$ value than b-b', due to both a steeper ice cliff and a much steeper backwasting ramp. The results show that ice cliffs with streams tend to have lower $\psi$ values as a result of steeper ramp angles for ice cliffs with similar pitch (Figure 6d-e), providing evidence that supraglacial

streams enhance the efficiency of ice cliff incision.

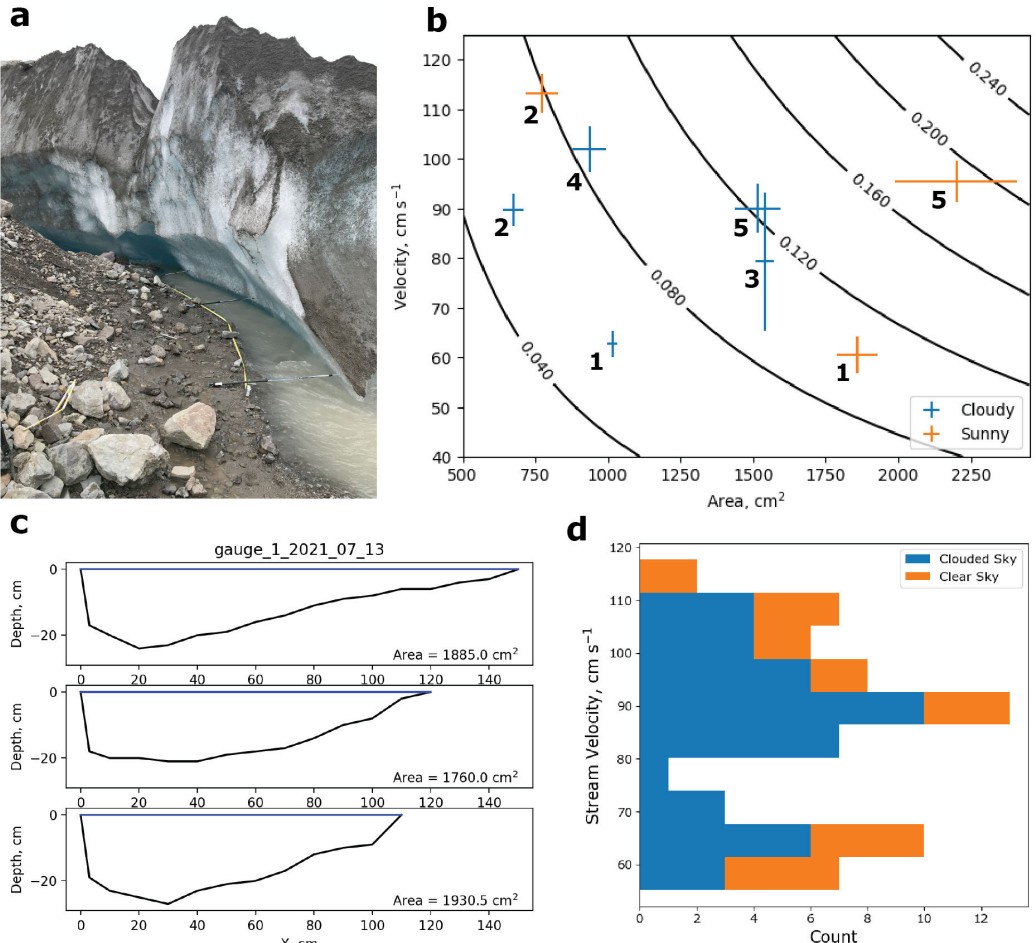

**Figure 7.** Results of stream discharge measurements. (a) Image of typical setup of discharge measurement including hiking poles ($\sim$1.4 m length) marking the flux gates and tape measure. (b) Measured velocity and cross-sectional area for all discharge measurements; plotted contours are the resultant discharge in m$^3$ s$^{-1}$. Data points are distinguished by measurement site (numerical labeling) and sunny/cloudy conditions (color). (c) Example of channel cross-sectional area measurements, from site 1 on July 13, 2021. (d) Histogram of measured velocity values from all 64 datapoints from 8 measurement sets.

## 4.4 Stream discharge

The cross-sectional area of individual stream discharge measurement gates ranged from 630-2480 cm$^2$, with stream widths between 57-150 cm and maximum depths up to 12-35 cm. The thalweg, or deepest part of the channel cross-section, was in nearly all cases located nearer to the channel wall contiguous with the base of ice cliffs (Figure 7c; Supplementary Figures S4-7). This is consistent with lateral stream meander towards ice cliff undercuts. Mean measured velocities ranged from 0.60-1.13 m s$^{-1}$, with standard deviation in measured values ranging from 0.03-0.13 m s$^{-1}$. The resultant flow rates were determined between 0.06 - 0.21 m$^3$ s$^{-1}$ (Figure 7).

The streams sampled in the discharge measurements tended to be nearly continuously ice-floored, with little-to-no sediment resting on the channel bottom. When observed, sediments tended to be small accumulations of fine gravels/coarse sands collected in hollows or as small highly mobile bedforms (ripples), or larger rocks protruding above the surface of the stream. These larger static rocks were avoided in discharge measurement location selection. The streams themselves tended to be highly turbid, with low visibility beyond 5-10 cm.

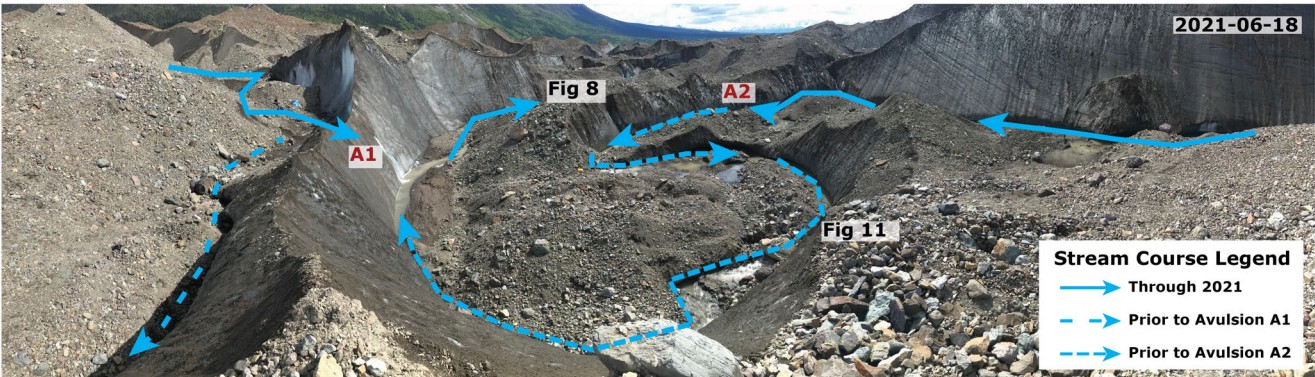

**Figure 8.** Perspective image of a large (∼40 m wide) topographic bowl bounded by a supraglacial stream meander undercutting associated ice cliffs. Observed stream courses and flow direction are displayed with the blue arrows. Dashed lines indicate stream courses abandoned following avulsions A1 (occurred sometime prior to June 18, 2021) and A2 (observed to occur on July 15, 2021). The locations of avulsions A1 and A2 are noted, as well as the locations of ice cliffs shown in Figures 9 and 12. Avulsion A2 resulted in stream flow directly into the page/downview of the camera.

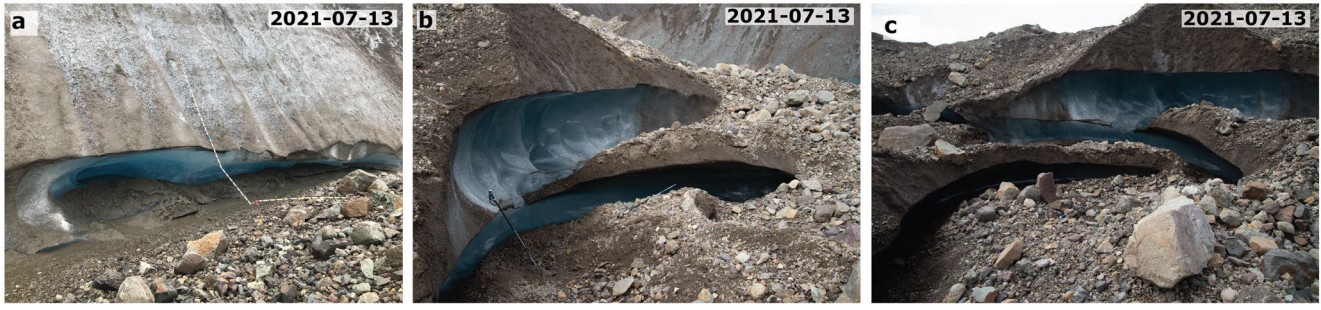

**Figure 9.** Examples of stream undercuts of ice cliffs observed after avulsions; ablation stake with 1.5 m long segments connected by a string and hiking pole ∼1.4 m in length for scale.

## 4.5   In situ geomorphic observations

In situ observations find supraglacial streams frequently at the base of ice cliffs, consistent with remote sensing results (Figures 4, 8). Streams undercut the ice cliffs through lateral meandering and incision. Abandoned stream channels reveal up to >2.5 m horizontal and >1 m vertical incision (Figure 9a-b). In the case of repeated incision by multiple episodes of stream meander action, up to 3.5 m total vertical incision was observed, a significant local contribution to the total area of ice cliffs (Figure 9c).

There are also observed examples in which the action of supraglacial runoff contributes to ice cliff generation. Small rivulets of runoff, generated by the melt from ice cliffs and/or catchments of debris-covered ice, produce local incision and meanders as they travel through the supraglacial debris layer. This small scale incision generates ice exposures on the order of centimeters to tens of centimeters in height which have the potential to grow into much larger scale ice cliffs. In one dramatic example, a time lapse camera used in fieldwork was knocked over by the generation and backwasting of an ice cliff that grew from 10 cm height cut by a rivulet to 1 m height and independent of the original runoff system within 38 days (Figure 10).

The impacts of two supraglacial stream avulsion events on ice cliffs were directly observed on Kennicott Glacier in the summers 2021 and 2022 (Figures 8, 11). These avulsions, i.e., rapid abandonments of the stream channel and formation of new channels, took place as a result of stream meanders in the same stream course system approaching each other from opposite directions, in manner similar to oxbow lake production in meandering river systems. As the stream meanders approach each other, the ice cliffs associated with the cut bank side of the meander meet, producing an ice fin with a sharp crest. Once the ice fin thins sufficiently the upstream meander breaks through and the stream avulsion occurs, abandoning the prior stream path between those two points. Rapid stream flow at the avulsion point leads to accelerated collapse of the ice fin. In some cases, stream undercutting can contribute to mechanical collapse of ice cliffs and fins as well.

Channels abandoned by stream avulsion events initially reveal the extent of stream incision under ice cliffs. However, the shape of the overhang eventually melts out and in many cases the ice cliff begins to be reburied under debris. This was observed for south-facing ice cliffs following avulsion A2 on July 15, 2021 (Figure 12).

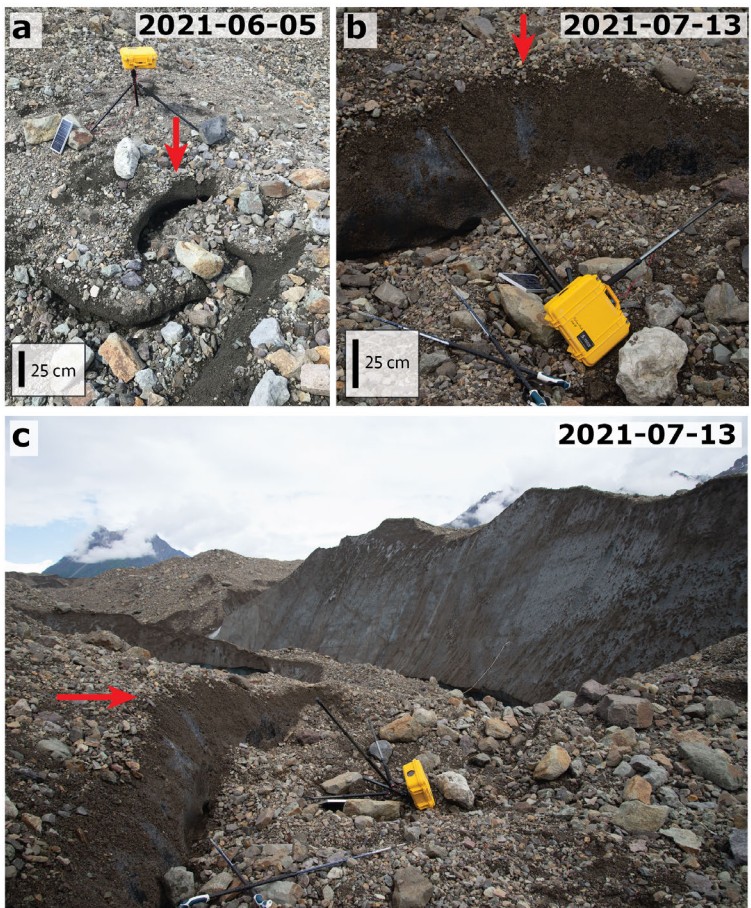

**Figure 10.** Field evidence for the role of hydrology in generating ice cliffs. (a) Time lapse camera observed on June 5, 2021. Red arrow indicates the crest of a short (∼10-20 cm tall) ice exposure associated with the incision of minor supraglacial drainage within the debris layer. The natural meander in drainage is expressed in the shape of the ice exposure. (b-c) By July 13, 2021, the ice exposure had grown substantially into a small ice cliff ∼1 m tall and several meters wide, backwasting through the topography and knocking over the time lapse camera. This also illustrates the need for careful evaluation of local terrain features when placing instruments on the surface of debris-covered glaciers.

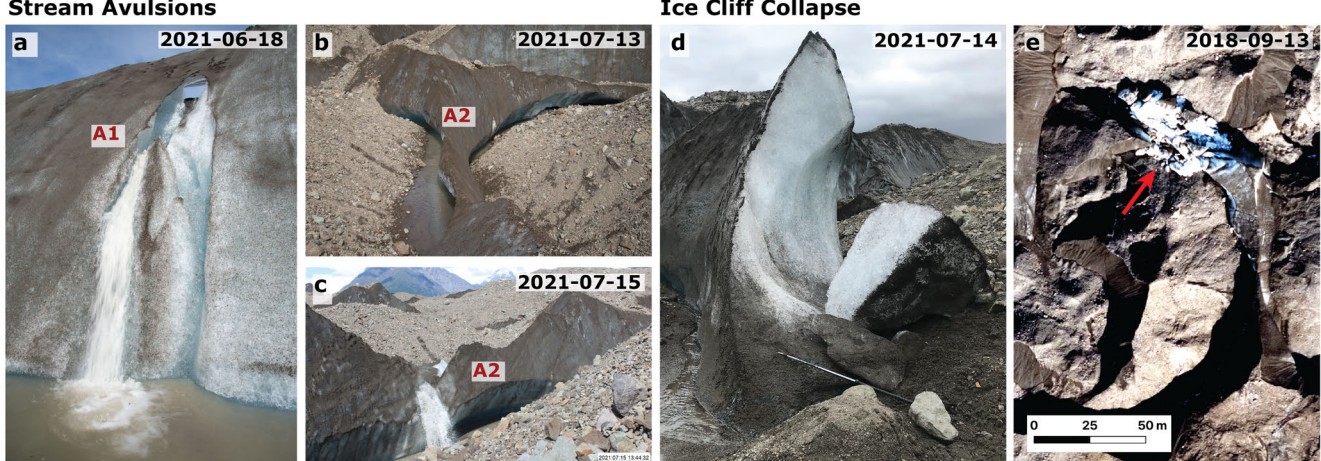

**Figure 11.** Examples of supraglacial stream avulsions and undercutting resulting in ice cliff collapse. (a) Avulsions A1 (occurred sometime prior to June 18, 2021), with stream flow boring through thinning ice cliff fin. (b-c) Avulsion A2 (observed to occur on July 15, 2021). Thinning of the ice cliff fin and lateral stream migration resulted in rapid ice cliff collapse and stream avulsion. (d-e) Observed ice cliff collapses associated with stream undercutting, (d) on the ground, and (e) observed in photogrammetry data.

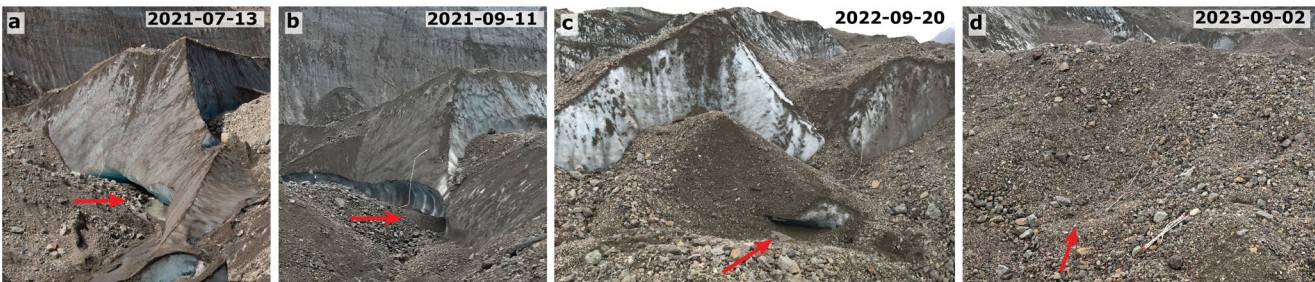

**Figure 12.** Observed reburial of south-facing ice cliff under debris following stream channel abandonment. (a) Ice cliff with active supraglacial stream at its base, providing undercutting via lateral meandering. Base of cliff is unburied. (b) After avulsion A2 (Figure 11), undercutting ceases and undercut overhang begins to melt out. (c) After complete melting of the undercut shape, the ice cliff begins to be reburied under debris. (d) Complete reburial of the former ice cliff area under debris 1 year after supraglacial stream avulsion.

## 5  Discussion

### 5.1  Geomorphic relationship between ice cliffs and streams

It is clear from the results of our DEM analysis that supraglacial streams and ice cliffs have a strong relationship in terms of spatial proximity and geomorphology. Linear to crescent-shaped cliffs are often observed with a supraglacial stream at their base or within tens of meters (Figure 4). Half of observed streams by length are within 10 m of ice cliffs and 32.7% of ice

cliffs have supraglacial streams directly at their base (Figure 5). Kneib et al. (2023) determined that 38.9% of ice cliffs in High Mountain Asia are stream-influenced, defined as being within 40 m of supraglacial streams. If we applied this broad threshold to our analysis, 91% of ice cliffs on Kennicott Glacier are stream-influenced. Conservatively, one third to one half of ice cliffs on Kennicott Glacier are stream-influenced. These cliffs typically follow the arc of supraglacial stream meanders on the outside of stream bends, analogous to cut banks in sedimentary river systems. This morphology is likely indicative of ice cliffs that have hydrological processes as their primary control.

Supraglacial streams act on the surface of the glacier through two processes: thermoerosional incision of the ice and the erosion/transport of supraglacial debris. Karlstrom et al. (2013) showed that because heat transfer between stream flow and ice channel walls is dependent on turbulence in a similar fashion to sediment transport, supraglacial streams produce meanders with a very similar planform geometry to alluvial rivers. This lateral as well as vertical thermoerosional incision allows streams to carve new ice cliffs into the glacier surface (Figure 10), as well as undercut and enhance the incision of existing ice cliffs (Figure 9). Indeed, we have shown evidence from our geomorphic model that the presence of streams measurably enhances the incision of ice cliffs into the glacier surface, aiding in ice cliff survival (Figure 6d). More pronounced lateral migration of streams under ice cliffs leads to more exposed ice and enhanced backwasting at the ice cliff base, as well as possible mechanical collapse (Figure 11). This is the mechanism by which stream channel meanders may control the morphology of ice cliffs, developing crescent-shaped cliffs on their outside bends.

Crescent cliffs adjacent to supraglacial stream systems often appear to be formerly associated with stream channel meanders, in a manner reminiscent of oxbow lakes adjacent to sedimentary river systems (Fisk, 1947). Observation of multiple stream avulsion events during 2021 fieldwork confirmed that stream abandonment of ice cliffs occurs, due to both meander pinch-off or lateral migration (Figures 11, 4). These streams leave behind ice cliffs with a morphology formerly influenced by stream action. Ice cliffs abandoned by streams lose the undercut morphology at their base and may be re-buried by debris, particularly if they are equator-facing (Figure 12; Buri and Pellicciotti (2018)). This observation further underlines the importance of streams to ice cliff survival.

If the vertical incision component of a supraglacial stream is sufficiently greater than the average local melt rate of sub-debris ice and ice cliffs, it may lead to the production of a cut-and-closure channel whereby the stream enters the englacial hydrological system (Gulley et al., 2009; Benn et al., 2009). This process becomes inevitable if the supraglacial stream reaches sufficient size. Most significant supraglacial streams observed during Kennicott Glacier fieldwork 2020-2023 eventually entered the glacier via a moulin or cut-and-closure channel. As debris thickness increases and sub-debris melt rates are reduced, relative stream incision increases and cut-and-closure channels are more easily produced by smaller streams; this is one of the major reasons that supraglacial hydrological networks tend to be smaller on debris-covered glaciers compared to bare glaciers (Miles et al., 2020). This process may reduce the importance of streams to ice cliff morphology for glaciers with thicker debris. Meanwhile the presence of englacial channels contributes to ice cliff formation upon their collapse, and the morphology of resultant ice cliffs will reflect that of the englacial stream course.

## 5.2 Stream erosion of sediments at the base of ice cliffs

In sediment-bedded streams, the ability of the stream to erode (mobilize from rest), transport (carry within the flow), or deposit
(drop out of the flow) clasts of a given size as a function of stream velocity has historically been described by a set of empirical
relationships illustrated by the Hjulstrom curve (Hjulström (1935); Supplementary Figure S8). Comparison of our discharge
measurement results to this diagram suggests the ability to erode debris clasts up to 1 cm in size and transport debris up to 10
cm in size.

However, because our supraglacial streams are generally bedded by ice and not sediment, the Hjulstrom diagram is likely to
be ill-suited for estimation of the debris erosion/mobilization threshold. If we assume a smooth ice-floored stream bed, we can
predict the onset of sediment transport (erosion) with a simple force balance. We may consider the drag force $F_{drag}$ required
to overcome the static frictional force $F_{friction}$ of a debris clast resting on the ice in a stream channel:

$$F_{drag} = \frac{AC_w \rho_{water} v_{flow}^2}{2} \tag{2}$$

$$F_{friction} = \mu F_N = \mu \rho_{rock} V_{rock} g \tag{3}$$

where A is the cross sectional area of the rock orthogonal to flow direction, $C_w$ is the drag coefficient, $\rho_{water}$ and $\rho_{rock}$ the
densities of water and rock, $v_{flow}$ the stream velocity, $\mu$ the static coefficient of friction (0.05 for ice near 0°C (Mills, 2008)),
$F_N$ the normal force, $V_{rock}$ the rock volume, and $g$ acceleration due to gravity. At the critical drag force, $F_{drag} = F_{friction}$,
we can determine the maximum size of rock of a certain shape that is mobilized by the flow.

We assumed a rock density of 2700 kg m$^{-3}$ and used a half-sphere ($C_w$ = 0.42) and a cube ($C_w$ = 1.05) (Hoerner, 1965)
as end member shapes to calculate the diameter of debris clasts our measured stream velocities are capable of mobilizing. We
found our measured velocities (0.6 - 1.13 m s$^{-1}$) capable of moving rocks up to 9-30 cm (half-sphere) and 14-51 cm (cube) in
diameter. These values match or exceed our measured channel depths.

These results are consistent with the fact that little static sediment accumulation is observed in the measured stream channels.
Where observed, static sediment size is typically the order of or larger than the stream depth, or it is gravel and sand collected
in hollows in the rough stream bed (our force balance assumes a smooth bed). Clasts larger than stream channel depth have
also been observed in motion in ice-bedded streams.

This analysis thus supports that supraglacial streams are highly effective at removing debris from the base of ice cliffs, which
may aid in preventing their reburial. The relative importance of debris erosion and transport in comparison to thermoerosional
undercutting of the ice cliff face is unknown.

## 5.3 Feedbacks and the role of ice dynamics in ice cliff evolution

Because ice cliffs contribute significantly to melt runoff, there is additionally the opportunity for a feedback loop between ice
cliffs and streams. More exposed ice cliff area leads to higher flow and incision rates with more power to create and maintain

ice cliffs. Greater debris thickness leads to a higher differential melt between sub-debris and ice cliff melt rates, increasing the relative importance of such a feedback. At the same time, lower sub-debris melt rates leads to significantly reduced runoff from catchment areas without ice cliffs and a reduced ability of runoff to generate new cliffs.

Thicker supraglacial debris may slow the development of new stream networks by inhibiting runoff. At the same time, the relative impact of existing supraglacial streams will be increased due to reduced sub-debris melt rates. This will lead to greater impacts on ice cliffs for the same stream size, but larger streams are progressively lost to cut-and-closure conduit formation as discussed in section 5.1. Larger clasts are less effectively removed by streams from the base of ice cliffs; they may induce meandering causing the stream to locally abandon the cliff or lead to deeper lateral undercutting. The presence of a sufficient volume of clasts larger than the stream can transport may lead to infilling of the stream channel and ice cliff reburial. Such abandoned stream channels, buried by debris, have been observed in the literature (Miles et al., 2020). Periods of higher flow due to exceptional melt, rainfall, and glacial outburst flooding could mobilize larger clasts and lead to accelerated reorganization of the supraglacial and englacial hydrological system.

While this study is focused on investigating the processes by which supraglacial streams control ice cliff formation and change, there are a number of other processes which also have major impacts on ice cliffs. Glacial hydrology more broadly can impact ice cliffs through other processes. The collapse of englacial conduits may lead to ice cliff production as a result of the exposure of former conduit walls (Benn et al., 2001; Mölg et al., 2020). Supraglacial ponds often are rimmed in part by ice cliffs (Watson et al., 2017a; Steiner et al., 2019), with the pond playing a key role in altering ice cliff surface energy balance and undercutting through enhanced subaqueous melt rates (Miles et al., 2016). Ice cliffs may also be generated through the modification of crevasse walls. Kneib et al. (2023) used geomorphic relationships to determine that supraglacial streams, crevasses, and supraglacial ponds are the primary controls on ice cliffs in High Mountain Asia, in order of decreasing importance.

On Kennicott Glacier supraglacial streams are prevalent across the majority of the active portion of the debris-covered tongue. The importance of streams are diminished (1) where active crevassing occurs and runoff is thus sent rapidly to the englacial system via moulins or (2) where stagnant ice results in flattened surface topography and the dominance of supraglacial ponds. At Kennicott Glacier, the ice becomes stagnant ∼2.5 km from its terminus at an elevation of ∼500 m.a.s.l. (Gardner et al. (2022); Anderson et al. (2021b), Figure 13); below this elevation streams are rare and ponds are dominant (Anderson et al., 2021b), while above this elevation streams are more common (Figure 1).

We also observed evidence that local patterns in ice dynamics plays a major role in ice cliff formation. Based on the ITS_LIVE velocity dataset derived from LANDSAT and other records (Gardner et al., 2018, 2022) we observed two episodes of ice flow acceleration, from 1988-1995 and 2018-2020. These occurred on the western half of the glacier from 2.5-7 km up valley of the terminus (Figure 13a, Supplementary Video 3). Velocities quadrupled in the area of interest for the most recent surge, from 23 m/yr (2014) to 98 m/yr (2019) (they roughly tripled for the previous surge, from 30 m/yr in 1986 to 95 m/yr in 1990). This "mini-surge" caused a minor localized advance and topographic bulge at the western glacier margin but had no impact on the glacier terminus. Field observations showed that the area coinciding with the strongest speed-up was heavily crevassed in 2019-2020 (Figure 13d), with freshly exposed ice cliff faces. These new ice cliffs, originating as crevasse walls,

can be identified by their morphology in our ice cliff inventory by their elongated linear morphology and tendency to occur in parallel sets (Figure 13c).

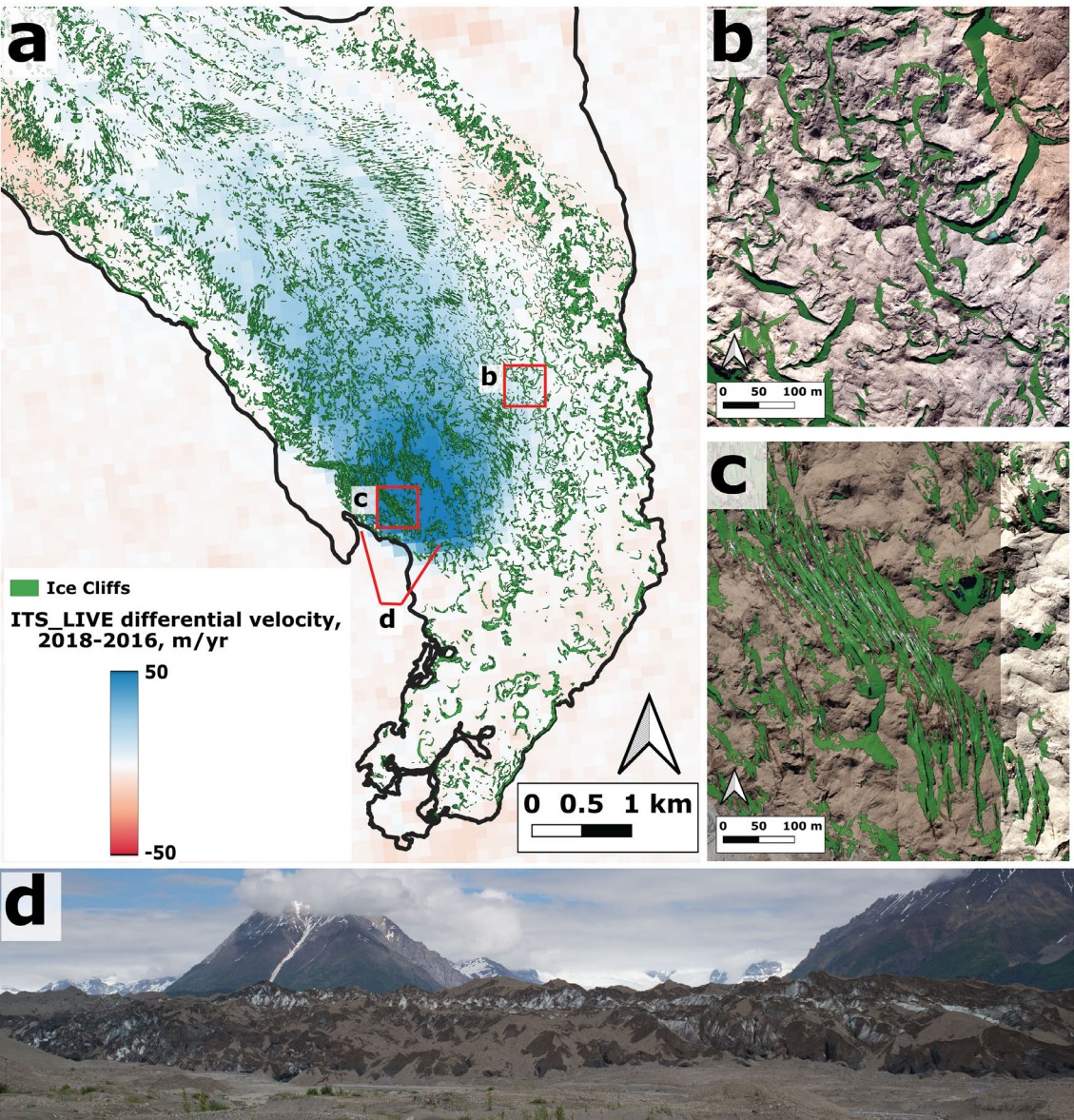

**Figure 13.** The impact of a localized surge-like event on ice cliff generation and morphology. (a) Kennicott Glacier terminus with mapped difference in ITS_LIVE surface velocities between 2018 and 2016; positive values on the western terminus delineate the location of the 2018 readvance. (b) Ice cliffs outside the speed-up area, dominantly exhibiting crescentic stream-influenced morphologies. (c) Ice cliffs in speed-up area where new crevasses are formed, exposing ice in repeated linear orientations. (d) View of the surge area in June 2020; exposed ice is clearly visible in ice cliffs formed from crevassing.

Thus for this area ice dynamics supersedes supraglacial hydrology as the dominant geomorphic driver of ice cliff formation and evolution. The crevassing and ice cliff formation related to the flow acceleration may be partly responsible for the higher percentage by area of ice cliff coverage in this study compared to that constrained in 2011 data by Anderson et al. (2021b) (14% vs 12%). Debris-covered glaciers which experience surges (Glasser et al., 2022) may exhibit ice dynamics as the primary control on ice cliff formation. Interestingly, Kneib et al. (2021) found that a surge which terminated upstream of debris-cover on Urdok glacier led to enhanced meltwater routing to the surface and thus enhanced ice cliff formation due to stream water incision of cryo-valleys. Dynamic, hydrologic, and other drivers of ice cliff change may be further linked through other complex feedbacks.

A number of other studies have noted a close geospatial linkage between supraglacial streams and ice cliffs on other glaciers around the world, including Zmuttgletscher (Mölg et al., 2020), Trakarding Glacier (Sato et al., 2021), and glaciers in High Mountain Asia (Kneib et al., 2023). Our work shows the processes by which streams contribute to ice cliff maintenance and production at Kennicott Glacier; these processes are likely universal and applicable to other debris-covered glaciers such as those listed above.

## 6 Conclusions

We have shown the importance of supraglacial streams in providing a feedback and control on ice cliff formation, dynamics, and survival on the debris-covered Kennicott Glacier. Streams running at the base of fully developed ice cliffs enhance the survivability of the ice cliff system through thermo-erosional incision of the ice, erosion of supraglacial debris which mass wastes to the base of the ice cliff, and undercutting of the ice cliff face. Undercutting of the ice face and erosion of debris prevents the ice cliff base from being re-buried by debris, while the undercut geometry increases exposed ice surface area available for melt. We find that 33% of ice cliffs are actively influenced by streams while 50% are in close proximity (<10 m) to streams.

Equator-facing ice cliffs are generally not expected to persist due to the effects of differential insolation-driven ablation leading them to be flattened and reburied (Buri and Pellicciotti, 2018). We observed that a south-facing ice cliff abandoned by a stream was reburied by debris within one year (Figure 12). We thus interpret that stream action at the base of ice cliffs is sufficient to maintain the cliff's survival, counteracting ablation-driven flattening.

Stream meanders also control the morphology of ice cliffs, producing curving, crescent, and sinuous ice cliff shapes on the outer edge of stream meanders. This morphology, highly reminiscent of the shape of oxbow lakes, may be a useful diagnostic to infer stream hydrology as the primary driver of a given ice cliff in remote sensing data. Meander cut off stream avulsions result in the demise of ice cliffs due to collapse at the avulsion point as well as reburial under debris where ice cliffs are abandoned by stream systems. The incision of streams into the debris-covered glacier surface may also produce fresh ice exposures that can grow into larger ice cliffs.

In this study we also observed clear evidence for the importance of ice dynamics in ice cliff formation and change. A localized surge-like flow acceleration between 2018 and 2020 correlated with a shift in ice cliff morphology from crescentic

stream-affected morphologies to repeated linear orientations indicating they originated as crevasses. Thus for this part of the glacier and this time period ice dynamics dominated over hydrology in driving ice cliff evolution.

Understanding the drivers of ice cliff formation and evolution will aid us in predicting how debris-covered glacier surfaces will change in the future, enhancing our understanding of melt, retreat, and discharge from these glaciers. Our study highlights specific drivers resulting from supraglacial hydrology as well as ice dynamics. In the future, landscape evolution models may be developed to predict the relative importance of these drivers and their feedbacks with other important factors not explored in this study, including differential melt, supraglacial ponds, and the collapse of englacial voids. Such work may ultimately strive to derive steady state ice cliff statistics for glaciers based on given climatic, glaciological, and debris-supply conditions.

*Data availability.*   Data products produced by this study are stored in a data repository at Petersen et al. (2023).

*Competing interests.*   We have no competing interests to declare.

*Author contributions.*   E. Petersen designed the study, led fieldwork data collection, carried out data analysis and drafted the manuscript. R. Hock was Principal Investigator of the project, and contributed to development, field work as well as manuscript editing. M. Loso provided the photogrammetry dataset and contributed greatly to geomorphologic analysis/interpretation and manuscript writing.

*Acknowledgements.*   The project was supported by NSF Award 1917536 (GLD). We thank field volunteers who made this work possible, including Cameron Markovsky, Julian Dann, Anna Thompson, Andrew Johnson, Nicole Trenholm, Brooke Kubby, Ruitang Yang, Kitrea Takata-Glushkoff, Maria Zeitz, Jason Geck, Harlan Loso, and Levi Williamson. We thank the Wrangell Mountains Center for their support during multiple field expeditions.

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
