# Peer review of "Stream hydrology controls on ice cliff evolution and survival on debris-covered glaciers"

_EGUsphere, 2023_

## Referee Comment (RC1)

The manuscript *Stream hydrology controls on ice cliff generation, evolution, and survival on debris-covered glaciers* by Peterson and co-authors presents a detailed geomorphological study of supraglacial streams on Kennicott Glacier, and their influence on the ice cliff characteristics and distribution at the surface of debris-covered glaciers. It combines detailed field observations, photogrammetric surveys and introduces a model representation of stream incision at the base of ice cliffs.

I found this manuscript particularly well written and interesting, as it presents a detailed analysis of the patterns and controls of ice cliffs on a glacier with a high concentration of supraglacial streams, a dense cliff distribution and relatively thin debris, which contrasts from previous studies which usually focused on glaciers with thicker debris with mostly pond-influenced cliffs. I really liked the idea of the geomorphic model introduced here, which is novel and links nicely cliff backwasting, stream incision and sediment transport. I still had some specific concerns related to some of the methods, and more general comments related to the discussion, that I hope the authors can address.

**General comments**

Cliff mapping: The approach described L68-73 is a bit vague and uses a lot of parameters that have unjustified values. There is a variety of methods that have been used to map ice cliffs in a (semi-)automated way from DEMs (Herreid and Pellicciotti, 2018) or optical images (Kneib et al., 2020; Kraaijenbrink et al., 2016; Anderson et al., 2021), and the choice of a different approach would therefore need to be well justified. As it is, it is very vague and unclear. Why not simply use the orthoimage? Was there no manual refinement of the outlines? What is the uncertainty of the mapping? Is there no effect from the very high spatial resolution that could lead to influence from large boulders for example? This is especially critical as 1/ it is not clear how much the mapping influences the analysis presented here and 2/ there is no discussion whatsoever of this automated mapping method. At the very least, it would be useful to compare the outlines of this approach with those from Anderson et al. (2021).

Choice of cliff profiles: The location of the profiles perpendicular to the cliffs appears to be pretty random. It would be useful to make a more systematic analysis for at least a few cliffs on how much the choice of the location of these profiles matters. I once tracked ice cliffs with time-lapse photogrammetry on a glacier in SE Tibet and made similar profiles, but realized that depending on the location of the profile along the cliff and its connection to the supraglacial stream, this could really change the surface profile and its evolution (Kneib et al., 2022).

Estimating thermo-erosional undercutting: I found the paragraph on the estimation of debris transport by the streams very interesting. However, it seems to me that the missing link in your conceptual model would be link between stream incision rate and ramp angle/characteristics. Would there not be a way to estimate this stream incision rate relative to the debris-covered ice? This appears to be quite a crucial element as I suspect that if the glacier had been more thickly debris-covered, the incision would have been too important and could have led to a cut-and-closure mechanism (Gulley et al., 2019).

Effects of dynamics on ice cliff behavior: I found the supplementary text S1 very interesting and would recommend including it in the main text. I also looked in a previous study at the influence of glacier speed-up on cliff distribution and characteristics, and there the speed-up led on the contrary to more stream-influenced cliffs, most likely as the changes in glacier dynamics led to more water being rerouted at the surface (Kneib et al., 2023). A link with glacier surges could also be relevant for this discussion (Glasser et al., 2022).

**Line-by-line comments**

L25: It would be good in this paragraph to add a few numbers on melt enhancement by ice cliffs (e.g. Miles et al., 2022), as well as on ice cliff density at the surface of glaciers. Specific studies looking into the role of ice cliffs on the debris-cover anomaly at the glacier scale could also be mentioned (e.g. Brun et al., 2018; Zhao et al., 2023)

L30: Not sure the term 'models of retreat' is appropriate. Would rather suggest 'debris-covered glacier melt models'.

L31: I would also note the work by Watson et al., 2016; 2017 and Westoby et al., 2020.

L33: another useful reference here would be the work by Bartlett et al. (2021).

L36: Remove 'Scott' Watson's first name in reference. This is a recurring error in the text. A similar problem occurs for the Garner et al. references.

L46: Here it is noteworthy that so far, most studies had so far focused on glaciers with thick debris cover where the main control on ice cliff evolution comes from the supraglacial ponds (Langtang, Khumbu, Lirung, Ngozumpa, Miage…). Thus the need for detailed work on the influence of stream hydrology.

Figure 1: Specify somewhere that the arrow indicates north (or put a N above). Also for the other figures. DEM and NPS acronyms need to be defined. Since the outline has been changed, could you also show the new outline?

L53: I find that a few details are lacking in this description. Especially related to debris thickness and flow velocities, since these are referred to later on.

L54: Could you add the coordinates of the glacier?

L57: specify melt rates 'of the debris-covered area'.

L63: specify if this is above the ground (more relevant here) or sea level. If it is sea level, please specify the altitude above the ground.

L67: same thing, mention m a.s.l.

L74: Why was a downsampling of the DEM necessary? Were some tests conducted to assess the influence of the DEM resolution on the mapped streams?

Figure 2: I believe there is no space at backwastes. Rather than simply stating that the velocity is negligible, can you be a bit more specific?

L105: In doing so, there is still a risk of having of lateral shift between the DEMs. This should be corrected for or at least quantified (Nuth and Kääb, 2011).

Figure 3: please remind the source of the orthophoto used.

Figure 5: these are impressive and very interesting results. However, as mentioned in the general comment, these profiles were selected manually and there be some subjectivity in this choice. A more systematic approach (more profiles, and several profiles per cliffs) would be needed to guarantee the strength of these results.

L153: something seems wrong with the syntax of this sentence.

Figure 6: Are there really no temperature measurements that could be exploited and could give a more quantitative idea compared to the qualitative 'cloudy/sunny' difference?

Figure 9: I don't see any scale in panel C. Could you perhaps add an estimation of the total cliff height in the image?

L200-202: this sounds more like a discussion statement

L234: this would be figure S5. Check numbering of supplementary figures.

L256: could this thermoerosional undercutting not be estimated?

**REFERENCES**

Anderson, L. S., Armstrong, W. H., Anderson, R. S., & Buri, P. (2021). Debris cover and the thinning of Kennicott Glacier, Alaska: in situ measurements, automated ice cliff delineation and distributed melt estimates. *The Cryosphere*, *15*, 265–282. https://doi.org/10.5194/tc-15-265-2021

Bartlett, O. T., Ng, F. S. L., & Rowan, A. v. (2021). Morphology and evolution of supraglacial hummocks on debris-covered Himalayan glaciers. *Earth Surface Processes and Landforms*, esp.5043. https://doi.org/10.1002/esp.5043

Brun, F., Wagnon, P., Berthier, E., Shea, J. M., Immerzeel, W. W., Kraaijenbrink, P. D. A., Vincent, C., Reverchon, C., Shrestha, D., & Arnaud, Y. (2018). Ice cliff contribution to the tongue-wide ablation of Changri Nup Glacier, Nepal, central Himalaya. *The Cryosphere*, *12*(11), 3439–3457. https://doi.org/10.5194/tc-12-3439-2018

Glasser, N. F., Quincey, D. J., & King, O. (2022). Changes in ice-surface debris, surface elevation and mass through the active phase of selected Karakoram glacier surges. *Geomorphology*, *410*, 108291. https://doi.org/10.1016/j.geomorph.2022.108291

Gulley, J. D., Benn, D. I., Screaton, E., & Martin, J. (2009). Mechanisms of englacial conduit formation and their implications for subglacial recharge. *Quaternary Science Reviews*, *28*(19–20), 1984–1999. https://doi.org/10.1016/j.quascirev.2009.04.002

Herreid, S., & Pellicciotti, F. (2018). Automated detection of ice cliffs within supraglacial debris cover. *The Cryosphere*, *12*, 1811–1829. https://doi.org/10.5194/tc-12-1811-2018

Kraaijenbrink, P. D. A., Meijer, S. W., Shea, J. M., Pellicciotti, F., de Jong, S. M., & Immerzeel, W. W. (2016). Seasonal surface velocities of a Himalayan glacier derived by automated correlation of unmanned aerial vehicle imagery. *Annals of Glaciology*, *57*(71), 103–113. https://doi.org/10.3189/2016AoG71A072

Kneib, M., Miles, E. S., Jola, S., Buri, P., Herreid, S., Bhattacharya, A., Watson, C. S., Bolch, T., Quincey, D., & Pellicciotti, F. (2020). Mapping ice cliffs on debris-covered glaciers using multispectral satellite images. *Remote Sensing of Environment*, 112201. https://doi.org/10.1016/j.rse.2020.112201

Kneib, M., Miles, E. S., Buri, P., Molnar, P., McCarthy, M., Fugger, S., & Pellicciotti, F. (2021). Interannual Dynamics of Ice Cliff Populations on Debris-Covered Glaciers from Remote Sensing Observations and Stochastic Modeling. *Journal of Geophysical Research: Earth Surface*, e2021JF006179. https://doi.org/10.1029/2021JF006179

Kneib, M., Miles, E. S., Buri, P., Fugger, S., McCarthy, M., Shaw, T. E., Chuanxi, Z., Truffer, M., Westoby, M. J., Yang, W., & Pellicciotti, F. (2022). Sub-seasonal variability of supraglacial ice cliff melt rates and associated processes from time-lapse photogrammetry. *The Cryosphere*, *16*(11), 4701–4725. https://doi.org/10.5194/tc-16-4701-2022

Nuth, C., & Kääb, A. (2011). Co-registration and bias corrections of satellite elevation data sets for quantifying glacier thickness change. *The Cryosphere*, *5*(1), 271–290. https://doi.org/10.5194/tc-5-271-2011

Watson, C. S., Quincey, D. J., Carrivick, J. L., & Smith, M. W. (2017). Ice cliff dynamics in the Everest region of the Central Himalaya. *Geomorphology*, *278*, 238–251. https://doi.org/10.1016/j.geomorph.2016.11.017

Watson, C. S., Quincey, D. J., Smith, M. W., Carrivick, J. L., Rowan, A. v, & James, M. R. (2017). *Quantifying ice cliff evolution with multi-temporal point clouds on the debris-covered Khumbu Glacier, Nepal*. https://doi.org/10.1017/jog.2017.47

Westoby, M. J., Rounce, D. R., Shaw, T. E., Fyffe, C. L., Moore, P. L., Stewart, R. L., & Brock, B. W. (2020). Geomorphological evolution of a debris-covered glacier surface. *Earth Surface Processes and Landforms*, *45*(14), 3431–3448. https://doi.org/10.1002/esp.4973

Zhao, C., Yang, W., Miles, E., Westoby, M., Kneib, M., Wang, Y., He, Z., & Pellicciotti, F. (2023). Thinning and surface mass balance patterns of two neighbouring debris-covered glaciers in the southeastern Tibetan Plateau. *The Cryosphere*, *17*(9), 3895–3913. https://doi.org/10.5194/tc-17-3895-2023

---

## Referee Comment (RC2)

The study by Petersen et al. "Stream hydrology controls on ice cliff generation, evolution, and survival on debris-covered glaciers" uses a number of methods to study ice cliff evolution and determine the importance of supraglacial streams on the longevity of ice cliffs. It is a well-written, detailed study that presents some interesting conclusions from a large amount of excellent data and observations. I think the results could be better utilised to support some of these conclusions, and I would like to see more detail in the Methods and Discussion sections, particularly in the discussion of the presented results.

General comments

1. While I appreciate the focus of the paper is on the impact of streams rather than supraglacial hydrology as a whole, I find the omission of any discussion of supraglacial ponds a little strange, particularly when there is mention on L157 that large cliffs are more generally associated with ponds near the terminus, and on L282 that ponds are dominant in this area. I think this needs to be at least acknowledged – even just a description of pond coverage in Section 2 would be helpful and a statement that ponds were not considered.

2. Throughout the Discussion, it would be good to see more discussion of your results – there are so many fantastic figures that are barely mentioned. For example, at L219 you could mention and discuss Figure 5, particularly panel D, which shows quite convincing evidence to support this statement but isn't currently mentioned at all. I would also like to see more development of the discussion in places, for example, at L266 – have you evidence for this point? Abandoned, debris-filled channels have been observed on debris-covered glaciers (e.g., Miles et al., 2020, 10.1016/j.earscirev.2020.103212), but streams can also meander around large debris clasts and might even undercut cliffs farther to flow around such obstructions. Periods of higher flow could also evacuate debris from channels and even reorganise the englacial/supraglacial system (e.g., during a flood event; Miles et al., 2018, 10.5194/tc-2018-152), creating new ice cliff faces.

3. One smaller point – in the title and conclusions (L290), I'm not convinced there is enough evidence to claim that streams control the generation of ice cliffs. For me, your results demonstrate the impact of streams on the dynamics and longevity of ice cliffs. The example in Section 4.4 (Figure 9) is a great complementary observation, but of ice cliff growth more than generation. I'd recommend removing "generation" from the title and conclusions and keeping the focus on the dynamics between existing ice cliffs and streams.

Specific comments

L25-38: I think these two paragraphs would read better if you swapped them around, so that you first discuss how ice cliffs are formed, then their importance on glacier surfaces. It would also save repetition, as you could remove the descriptive phrase in L25 ("where the glacier surface...").

L25: Give some values to demonstrate that melt from ice cliffs is "significant".

L29: Rowan et al. (2021, 10.1029/2020JF005761) found that ice cliffs only contributed a small proportion of the debris-cover anomaly on Khumbu Glacier.

L36 and throughout: Remove Scott Watson's first name (Scott) from references in-text and the bibliography.

Figure 1: Please consider using a colour-blind friendly colour scheme. It's also quite difficult to spot the red lines for cliff-ramp profiles in panel C – perhaps making the background more transparent, removing the black outlines from the other features, and making these lines bolder (and darker, perhaps black) would help.

Figure 1B: There are less cliffs (and streams?) at the terminus of the glacier, with an almost abrupt transition between there and the denser ice cliff coverage upglacier. It would be interesting to see a mention of this in the Discussion – is something preventing ice cliffs persisting at the terminus? The velocity map in Supplementary Figure 6 could be relevant (and isn't currently mentioned anywhere in the text, though the caption contains an interesting interpretation that could also be developed in the Discussion).

L58: A brief sentence on the coverage/number of supraglacial streams over the glacier surface would help the reader evaluate how accurate the DEM-predicted stream and cliff prevalence at this point in the paper (there seems to be a lot of both – is this the case? Are the streams all of a similar size or can only a proportion be seen from the glacier surface? How many are water-filled and how many are remnant channels?)

L65: Where is this airport? Can it be shown in Figure 1 or a distance and direction from the glacier given?

L68-73: Why were these angles chosen? Can some justification and/or references be given? I think the 50° on L72 should be 30°, as anything shallower is more likely to have debris? Was the aim only to sample completely ice-free cliffs? What about cliffs with a thin layer of very fine sediment deposited by melt rivulets?

L85: More information is needed here – are these total stream lengths and cliff areas or 2 m sections? Was the distance calculated using centre points or the shortest possible distance between features?

Figure 2C: Please consider using a colour-blind friendly colour scheme and/or line patterns.

L91: Backwasting ramps may not have been identified in the literature, but your schematic and model of ice cliff backwasting has a lot of similarities with that presented for ice sails by Evatt et al. (2017, 10.1017/jog.2017.72) and I think a brief comparison should be included.

L129: Can you give the uncertainty of these velocity measurements, or at least an idea of the variation (the standard deviation, for example)? It would also be worth noting that due to the time of day measurements were taken at, the velocities are likely near to the maximum daily velocity (and, later on, thus also the maximum potential for debris transport).

L139: I can understand the logic behind the methods used to delineate stream channels and cliffs, but I do think there needs to be some evaluation of the method and/or consideration (e.g., at the very start of the Discussion section) of how the results compare to what is actually seen on the glacier surface – from field observations or from the orthophoto. Can you give a rough idea of how many are water-filled vs. abandoned?

L142-146: Are these statistics for full channel lengths or the 2 m sections described in L85? A reminder here of how these were calculated would allow the reader to correctly understand the data and interpretations.

L149-150: The final clause of this sentence in an interpretation and should really be in the Discussion section.

L161: Is this stream location also confirmed by the field observations? Does it contain meltwater or has it been abandoned (i.e., active or historical undercutting)?

L168 and throughout (incl. figures and supplements): The journal recommends using SI units, though perhaps there is good reason for using cm?

L190-202: These are interpretations and should really come in the Discussion section.

L210: It could be interesting to present and discuss these statistics in two categories: 1) cliffs where the stream is immediately adjacent and actively influencing the cliff; 2) cliffs where the stream is farther away and has historically influenced the cliff. This could allow for an analysis over time of how the ice cliff profile changes as a stream moves farther away from it, though that's probably beyond the scope of this paper!

L261: Why an increased debris thickness? General debris-covered glacier trends? If so, give references.

L269: Reference Benn et al. (2001, 10.3189/172756501781831729) for ice cliff formation from collapse of englacial conduits – and at L35 in the Introduction.

L274: Was there a difference according to ice cliff size for the order of importance of these influences?

L283: Where are cliffs more common and is there a variation in size? What could be inferred about ice cliff longevity on the basis of this spatial variation in supraglacial hydrological features?

---

## Author Comment (AC1)

Response to Reviewers

Reviewers' comments in Blue
Response in Black

Comments from Marin Kneib:

**General comments**

Cliff mapping: The approach described L68-73 is a bit vague and uses a lot of parameters that have unjustified values. There is a variety of methods that have been used to map ice cliffs in a (semi-)automated way from DEMs (Herreid and Pellicciotti, 2018) or optical images (Kneib et al., 2020; Kraaijenbrink et al., 2016; Anderson et al., 2021), and the choice of a different approach would therefore need to be well justified. As it is, it is very vague and unclear. Why not simply use the orthoimage? Was there no manual refinement of the outlines? What is the uncertainty of the mapping? Is there no effect from the very high spatial resolution that could lead to influence from large boulders for example? This is especially critical as 1/ it is not clear how much the mapping influences the analysis presented here and 2/ there is no discussion whatsoever of this automated mapping method. At the very least, it would be useful to compare the outlines of this approach with those from Anderson et al. (2021).

The comparison with the outlines from Anderson et al. (2021) is not expected to be useful in terms of method validation due to the 10 year gap between the dataset. Anderson et al. (2021) uses a July 2009 WorldView image for their ice cliff delineation, whereas we use a September 2019 photogrammetry dataset. Changes in ice cliff coverage and morphology could certainly be expected, particularly in light of the readvance mentioned in the supplementary text S1. However, I do think that this comparison would be useful in terms of ice cliff change, so will include some more explicit comparison there.

What we can do is make a comparison of the auto-generated ice cliff outlines using our method to manually mapped ice cliff outlines on a sub-sample of the study area. The manual mapping will use the orthophoto from the same dataset, making the validation accurate. The area of the Kennicott Glacier's debris-covered terminus will be sliced into 500 m x 500 m grids and 15 grid cells will be randomly selected to manually map ice cliffs using the orthophoto. These results will then be compared with the results of automated cliff mapping, determining the area percentage of false positive ice cliff identifications and false negative missed identifications. This will provide us an estimate of the uncertainty of the auto mapping.

We will also work to make the methods section clearer and develop text to further discuss this methods' strengths/weaknesses and results. We see an advantage to the relative simplicity of our method and the ability to apply it to datasets such as LiDAR or ArcticDEM where no companion imagery is provided. In addition, it is not sensitive to variable lighting conditions or variable albedo of ice cliffs (e.g. cliffs that are cleared of fine grained debris and thus high albedo).

There is no manual refinement of cliff outlines. There may indeed be false positive ice cliff identification due to the edges of large boulders, however these false positives are expected to be relatively small in area. Thus it is not expected that this would have a major effect on the analysis presented pertaining to ice cliff-stream interactions.

More broadly, the analysis of ice cliff-stream interactions using the geomorphic model includes direct visual confirmation that automatically mapped ice cliffs are indeed true ice cliffs.

Choice of cliff profiles: The location of the profiles perpendicular to the cliffs appears to be pretty random. It would be useful to make a more systematic analysis for at least a few cliffs on how much the choice of the location of these profiles matters. I once tracked ice cliffs with time-lapse photogrammetry on a glacier in SE Tibet and made similar profiles, but realized that depending on the location of the profile along the cliff and its connection to the supraglacial stream, this could really change the surface profile and its evolution (Kneib et al., 2022).

This is a good suggestion. As shown in Figure 5, the selection of location for the profile can make a huge difference in the extreme case of a stream undercutting one section of the cliff but not another. I'm sure if for example the stream had only recently begun to undercut the cliff on one section that its signal would be much less pronounced. This sensitivity analysis (extracting many profiles along the length of the cliff and assessing the difference in results) is something we will do for 2-4 cliffs with/without undercutting streams and present the results in supplementary material.

Estimating thermo-erosional undercutting: I found the paragraph on the estimation of debris transport by the streams very interesting. However, it seems to me that the missing link in your conceptual model would be link between stream incision rate and ramp angle/characteristics. Would there not be a way to estimate this stream incision rate relative to the debris-covered ice? This appears to be quite a crucial element as I suspect that if the glacier had been more thickly debris- covered, the incision would have been too important and could have led to a cut-and-closure mechanism (Gulley et al., 2019).

Doing a proper estimation of stream incision rate does not seem trivial, and is beyond the scope of this paper. To do so from physical principles would require some treatment of turbulent heat fluxes from the flow (Karlstrom et al., 2013). To do so in the context of the geomorphic model would require deconvolution from other relevant parameters such as ice cliff incision efficiency/debris reburial of the ice cliff base.

We agree that if the glacier is more thickly debris-covered, incision would lead to cut-and-closure. In fact, it seems like the size of streams on Kennicott glacier is already inherently limited by this effect, as any larger stream network we observed inevitably transitions into an englacial system or drops into a moulin. Increasing debris thickness will then decrease the maximum stream size limit that already exists. Cut-and-closure channels may then provide a method of ice cliff formation as they melt out (Benn et al. 2001). I will at least include this in new language in the discussion section.

Effects of dynamics on ice cliff behavior: I found the supplementary text S1 very interesting and would recommend including it in the main text. I also looked in a previous study at the influence of glacier speed-up on cliff distribution and characteristics, and there the speed-up led on the contrary to more stream-influenced cliffs, most likely as the changes in glacier dynamics led to more water being rerouted at the surface (Kneib et al., 2023). A link with glacier surges could also be relevant for this discussion (Glasser et al., 2022).

Very interesting! I will include a reference to that study in the discussion. I'm also glad that you found supplementary text S1 valuable, and will accordingly relocate this text and the figure to the appropriate location in the discussion section where it is referenced.

**Line-by-line comments**

L25: It would be good in this paragraph to add a few numbers on melt enhancement by ice cliffs (e.g. Miles et al., 2022), as well as on ice cliff density at the surface of glaciers. Specific studies looking into the role of ice cliffs on the debris-cover anomaly at the glacier scale could also be mentioned (e.g. Brun et al., 2018; Zhao et al., 2023)

I've added a few sentences, as also recommended by the second reviewer, and included the suggested references. As a part of this, some mention of the ice cliff density is also injected.

L30: Not sure the term 'models of retreat' is appropriate. Would rather suggest 'debris-covered glacier melt models'.

Agreed that this is better wording. Changed.

L31: I would also note the work by Watson et al., 2016; 2017 and Westoby et al., 2020. L33: another useful reference here would be the work by Bartlett et al. (2021).

Thank you for these reference suggestions; I've added them in the text.

L36: Remove 'Scott' Watson's first name in reference. This is a recurring error in the text. A similar problem occurs for the Garner et al. references.

This error has been fixed with regards to "Scott" Watson. We don't have any "Garner et al." references.

L46: Here it is noteworthy that so far, most studies had so far focused on glaciers with thick debris cover where the main control on ice cliff evolution comes from the supraglacial ponds (Langtang, Khumbu, Lirung, Ngozumpa, Miage...). Thus the need for detailed work on the influence of stream hydrology.

This is a great point!

Figure 1: Specify somewhere that the arrow indicates north (or put a N above). Also for the other figures. DEM and NPS acronyms need to be defined. Since the outline has been changed, could you also show the new outline?

We will make the suggested clarifications to clarify the North arrow and mentioned acronyms. And yes, we have an updated glacier outline we can place into this map as well as in Figure S7 (to be moved to the main text in accordance with one of your major comments).

L53: I find that a few details are lacking in this description. Especially related to debris thickness and flow velocities, since these are referred to later on.

We've updated this text, putting in some values on flow velocity and debris thickness.

L54: Could you add the coordinates of the glacier? L57: specify melt rates 'of the debris-covered area'.

We've added the center coordinates of the glacier as reported in the Randolph Glacier Inventory, as well as sub-debris melt rates.

L63: specify if this is above the ground (more relevant here) or sea level. If it is sea level, please specify the altitude above the ground.

L67: same thing, mention m a.s.l.

These are above sea level; we've clarified.

L74: Why was a downsampling of the DEM necessary? Were some tests conducted to assess the influence of the DEM resolution on the mapped streams?

We downsampled the DEM in order to reduce computation time. We anticipate some loss of fine detail when scaling from 0.25 to 2 m resolution, but that relationships between streams and ice cliffs (at the meters to 10s of meters scale) are not expected to significantly impacted by the downsampling used.

Figure 2: I believe there is no space at backwastes. Rather than simply stating that the velocity is negligible, can you be a bit more specific?

We've removed the space from backwastes, making it consistent with the use of "backwasting" through the rest of the manuscript text. We've also made it clear that the velocity is negligible in the sense that the ITS_LIVE derived values in the vicinity of the ice cliff are comparable to off-glacier velocities.

L105: In doing so, there is still a risk of having of lateral shift between the DEMs. This should be corrected for or at least quantified (Nuth and Kääb, 2011).

We will perform a quantification of error due to horizontal shift by comparing DEM results for the location of a static moraine crest.

Figure 3: please remind the source of the orthophoto used.

We now appropriately reference the data source as well as the methods section where it is described.

Figure 5: these are impressive and very interesting results. However, as mentioned in the general comment, these profiles were selected manually and there be some subjectivity in this choice. A more systematic approach (more profiles, and several profiles per cliffs) would be needed to guarantee the strength of these results.

We find this criticism to be fair. As mentioned in the response to the general comment, we are willing to do some sensitivity analysis on cliff profile selection and its impact on our results.

L153: something seems wrong with the syntax of this sentence.

We reworded this sentence.

Figure 6: Are there really no temperature measurements that could be exploited and could give a more quantitative idea compared to the qualitative 'cloudy/sunny' difference?

Temperature measurements are available; however we did not find it necessary to leverage these in this study as we are not attempting to predict stream discharge via melt modeling. The use of clouded vs. clear sky days is simply to show some sense of day-to-day variability in stream discharge at the same locations under differing meteorological conditions.

Figure 9: I don't see any scale in panel C. Could you perhaps add an estimation of the total cliff height in the image?

Panel C is a different perspective on the same ice cliff shown in b; the red arrow indicates the same location in both frames. We did not place a scale in panel C as there is significant photographic foreshortening and thus any scale bar may be deceptive. However, we will be more clear in the caption about this as well as being explicit about the scale of the ice cliff.

L200-202: this sounds more like a discussion statement

We moved this to the discussion accordingly.

L234: this would be figure S5. Check numbering of supplementary figures.

In our most updated version this is Figure S6. I will double check these references before resubmission.

L256: could this thermoerosional undercutting not be estimated?

Estimation of thermoerosional undercutting is not trivial as it requires determination of turbulent heat fluxes at the water-ice interface (Karlstrom et al. 2013).

**REFERENCES**

Anderson, L. S., Armstrong, W. H., Anderson, R. S., & Buri, P. (2021). Debris cover and the thinning of Kennicott Glacier, Alaska: in situ measurements, automated ice cliff delineation and distributed melt estimates. *The Cryosphere*, *15*, 265–282. https://doi.org/10.5194/tc-15- 265- 2021

Bartlett, O. T., Ng, F. S. L., & Rowan, A. v. (2021). Morphology and evolution of supraglacial hummocks on debris-covered Himalayan glaciers. *Earth Surface Processes and Landforms*, esp.5043. https://doi.org/10.1002/esp.5043

Brun, F., Wagnon, P., Berthier, E., Shea, J. M., Immerzeel, W. W., Kraaijenbrink, P. D. A., Vincent, C., Reverchon, C., Shrestha, D., & Arnaud, Y. (2018). Ice cliff contribution to the tongue-wide ablation of Changri Nup Glacier, Nepal, central Himalaya. *The Cryosphere*, *12*(11), 3439–3457. https://doi.org/10.5194/tc-12-3439-2018

Glasser, N. F., Quincey, D. J., & King, O. (2022). Changes in ice-surface debris, surface elevation and mass through the active phase of selected Karakoram glacier surges. *Geomorphology*, *410*, 108291. https://doi.org/10.1016/j.geomorph.2022.108291

Gulley, J. D., Benn, D. I., Screaton, E., & Martin, J. (2009). Mechanisms of englacial conduit formation and their implications for subglacial recharge. *Quaternary Science Reviews*, *28*(19–20), 1984–1999. https://doi.org/10.1016/j.quascirev.2009.04.002

Herreid, S., & Pellicciotti, F. (2018). Automated detection of ice cliffs within supraglacial debris cover. *The Cryosphere*, *12*, 1811–1829. https://doi.org/10.5194/tc-12-1811-2018

Kraaijenbrink, P. D. A., Meijer, S. W., Shea, J. M., Pellicciotti, F., de Jong, S. M., & Immerzeel, W. W. (2016). Seasonal surface velocities of a Himalayan glacier derived by automated correlation of unmanned aerial vehicle imagery. *Annals of Glaciology*, *57*(71), 103–113. https://doi.org/10.3189/2016AoG71A072

Kneib, M., Miles, E. S., Jola, S., Buri, P., Herreid, S., Bhattacharya, A., Watson, C. S., Bolch, T., Quincey, D., & Pellicciotti, F. (2020). Mapping ice cliffs on debris-covered glaciers using multispectral satellite images. *Remote Sensing of Environment*, 112201. https://doi.org/10.1016/j.rse.2020.112201

Kneib, M., Miles, E. S., Buri, P., Molnar, P., McCarthy, M., Fugger, S., & Pellicciotti, F. (2021). Interannual Dynamics of Ice Cliff Populations on Debris-Covered Glaciers from Remote Sensing Observations and Stochastic Modeling. *Journal of Geophysical Research: Earth Surface*, e2021JF006179. https://doi.org/10.1029/2021JF006179

Kneib, M., Miles, E. S., Buri, P., Fugger, S., McCarthy, M., Shaw, T. E., Chuanxi, Z., Truffer, M., Westoby, M. J., Yang, W., & Pellicciotti, F. (2022). Sub-seasonal variability of supraglacial ice cliff melt rates and associated processes from time-lapse photogrammetry. *The Cryosphere*, *16*(11), 4701–4725. https://doi.org/10.5194/tc-16-4701-2022

Nuth, C., & Kääb, A. (2011). Co-registration and bias corrections of satellite elevation data sets for quantifying glacier thickness change. *The Cryosphere*, *5*(1), 271–290. https://doi.org/10.5194/tc-5-271-2011

Watson, C. S., Quincey, D. J., Carrivick, J. L., & Smith, M. W. (2017). Ice cliff dynamics in the Everest region of the Central Himalaya. *Geomorphology*, *278*, 238–251. https://doi.org/10.1016/j.geomorph.2016.11.017

Watson, C. S., Quincey, D. J., Smith, M. W., Carrivick, J. L., Rowan, A. v, & James, M. R. (2017). *Quantifying ice cliff evolution with multi-temporal point clouds on the debris-covered Khumbu Glacier, Nepal.* https://doi.org/10.1017/jog.2017.47

Westoby, M. J., Rounce, D. R., Shaw, T. E., Fyffe, C. L., Moore, P. L., Stewart, R. L., & Brock, B. W. (2020). Geomorphological evolution of a debris-covered glacier surface. *Earth Surface Processes and Landforms*, *45*(14), 3431–3448. https://doi.org/10.1002/esp.4973

Zhao, C., Yang, W., Miles, E., Westoby, M., Kneib, M., Wang, Y., He, Z., & Pellicciotti, F. (2023). Thinning and surface mass balance patterns of two neighbouring debris-covered glaciers in the southeastern Tibetan Plateau. *The Cryosphere*, *17*(9), 3895–3913. https://doi.org/10.5194/tc-17-3895-2023

Reviewer 2 Comments:

The study by Petersen et al. "Stream hydrology controls on ice cliff generation, evolution, and survival on debris-covered glaciers" uses a number of methods to study ice cliff evolution and determine the importance of supraglacial streams on the longevity of ice cliffs. It is a well-written, detailed study that presents some interesting conclusions from a large amount of excellent data and observations. I think the results could be better utilised to support some of these conclusions, and I would like to see more detail in the Methods and Discussion sections, particularly in the discussion of the presented results.

General comments

1. While I appreciate the focus of the paper is on the impact of streams rather than supraglacial hydrology as a whole, I find the omission of any discussion of supraglacial ponds a little strange, particularly when there is mention on L157 that large cliffs are more generally associated with ponds near the terminus, and on L282 that ponds are dominant in this area. I think this needs to be at least acknowledged – even just a description of pond coverage in Section 2 would be helpful and a statement that ponds were not considered.

Good point. We tried to acknowledge ponds adequately in the discussion, but we can include ponds in the study site description as well.

2. Throughout the Discussion, it would be good to see more discussion of your results – there are so many fantastic figures that are barely mentioned. For example, at L219 you could mention and discuss Figure 5, particularly panel D, which shows quite convincing evidence to support this statement but isn't currently mentioned at all. I would also like to see more development of the discussion in places, for example, at L266 – have you evidence for this point? Abandoned, debris-filled channels have been observed on debris-covered glaciers (e.g., Miles et al., 2020, 10.1016/j.earscirev.2020.103212), but streams can also meander around large debris clasts and might even undercut cliffs farther to flow around such obstructions. Periods of higher flow could also evacuate debris from channels and even reorganise the englacial/supraglacial system (e.g., during a flood event; Miles et al., 2018, 10.5194/tc-2018-152), creating new ice cliff faces.

Thank you for the suggestion. We can certainly draw more explicit references to the figures and results in the discussion as suggested.

The point raised in L266 could certainly be developed further. It is based on the idea that when the stream channel is filled in it cannot prevent the ice cliff from being reburied at its base. In addition, a thicker debris layer requires deeper incision to initiate ice cliff formation.

These are also interesting points raised about channel fill and flow variability that we will address/cite in the updated manuscript.

3. One smaller point – in the title and conclusions (L290), I'm not convinced there is enough evidence to claim that streams control the generation of ice cliffs. For me, your results demonstrate the impact of streams on the dynamics and longevity of ice cliffs. The example in Section 4.4 (Figure 9) is a great complementary observation, but of ice cliff growth more than generation. I'd recommend removing "generation" from the title and conclusions and keeping the focus on the dynamics between existing ice cliffs and streams.

We agree, you are correct that our results primarily demonstrate stream impacts on existing cliffs rather than on their generation. We can remove "generation" from the title and primary conclusions, but we will keep it in the discussion.

Specific comments

L25-38: I think these two paragraphs would read better if you swapped them around, so that you first discuss how ice cliffs are formed, then their importance on glacier surfaces. It would also save repetition, as you could remove the descriptive phrase in L25 ("where the glacier surface…").

I disagree on the swapping of paragraphs. Here I see the natural narrative flow as "Importance of Ice Cliffs" > "Ice cliff drivers" > "Supraglacial streams as drivers of ice cliffs specifically." I do however see the problem with the repetition, so I have changed my wording on L25 to "Exposed ice in ice cliffs contributes significantly to melt on debris covered glacier surfaces…"

L25: Give some values to demonstrate that melt from ice cliffs is "significant".

We have expanded upon the cited articles to be explicit about the values they present on melt from ice cliffs. This has also been informed by suggestions by Reviewer #1.

L29: Rowan et al. (2021, 10.1029/2020JF005761) found that ice cliffs only contributed a small proportion of the debris-cover anomaly on Khumbu Glacier.

We will cite this study and include more of the updated discussion on the debris cover anomaly.

L36 and throughout: Remove Scott Watson's first name (Scott) from references in-text and the bibliography.

This error has been fixed.

Figure 1: Please consider using a colour-blind friendly colour scheme. It's also quite difficult to spot the red lines for cliff-ramp profiles in panel C – perhaps making the background more transparent, removing

the black outlines from the other features, and making these lines bolder (and darker, perhaps black) would help.

We will rework this figure for better readability.

Figure 1B: There are less cliffs (and streams?) at the terminus of the glacier, with an almost abrupt transition between there and the denser ice cliff coverage upglacier. It would be interesting to see a mention of this in the Discussion – is something preventing ice cliffs persisting at the terminus? The velocity map in Supplementary Figure 6 could be relevant (and isn't currently mentioned anywhere in the text, though the caption contains an interesting interpretation that could also be developed in the Discussion).

You are right, we can develop this further in the discussion. It was mentioned briefly in line 277 the stagnant toe of the glacier with lower surface slope preventing the development of supraglacial streams. The lack of glacier flow as well as supraglacial streams removes two drivers of ice cliff formation or maintenance (crevasses and stream incision), which would explain the lower cliff density.

L58: A brief sentence on the coverage/number of supraglacial streams over the glacier surface would help the reader evaluate how accurate the DEM-predicted stream and cliff prevalence at this point in the paper (there seems to be a lot of both – is this the case? Are the streams all of a similar size or can only a proportion be seen from the glacier surface? How many are water-filled and how many are remnant channels?)

There is no prior work describing supraglacial streams on the glacier surface. We can include more in the discussion section on ground observations, however these are not systematic. The ice cliffs and streams are indeed numerous; this is a very large glacier. Streams vary greatly in size and stage of channel development and are incredibly dynamic. New streams are often observed without a well-developed channel, while well-developed active and abandoned channels are likewise often observed. Inactive abandoned channels are not expected to persist on the glacier surface for long due to melt and reburial by surface debris (e.g. Figure 11).

L65: Where is this airport? Can it be shown in Figure 1 or a distance and direction from the glacier given?

We will indicate the distance to the airport from the glacier in the text (~1.5 km) as well as airport coordinates.

L68-73: Why were these angles chosen? Can some justification and/or references be given? I think the 50° on L72 should be 30°, as anything shallower is more likely to have debris? Was the aim only to sample completely ice-free cliffs? What about cliffs with a thin layer of very fine sediment deposited by melt rivulets?

I think there is some confusion here as to our method. To be clear, 30 degrees is what we are using as the initial cutoff based on the debris angle of repose. However, this leads to false identification of many steep debris cone/moraine slopes on the glacier surface as ice cliffs. These debris-covered slopes tend to have their slope values clustered close to the angle of repose, with no steep values. The grand majority of well-developed ice cliffs on the other hand tend to have steep sections in excess of 50 degrees, as well as sections as low as 30 degrees. Hence the 50 degree value is a cut-off for the MAXIMUM slope

value for a contiguous candidate ice cliff shape, and is used as a method to reduce false positives. Many of our identified ice cliffs using this method have average slopes in the 30-something degrees range and are, as you describe, typical ice cliffs with a thin layer of fine sediment. We developed this method as described in the text through visual confirmation of ice cliff identifications.

L85: More information is needed here – are these total stream lengths and cliff areas or 2 m sections? Was the distance calculated using centre points or the shortest possible distance between features?

We will word this section more clearly. This is for 2 m sections, thus providing the shortest possible distance between features.

Figure 2C: Please consider using a colour-blind friendly colour scheme and/or line patterns.

We will rework this figure accordingly.

L91: Backwasting ramps may not have been identified in the literature, but your schematic and model of ice cliff backwasting has a lot of similarities with that presented for ice sails by Evatt et al. (2017, 10.1017/jog.2017.72) and I think a brief comparison should be included.

We will make note of the Evatt et al. (2017) model for ice cliffs.

L129: Can you give the uncertainty of these velocity measurements, or at least an idea of the variation (the standard deviation, for example)? It would also be worth noting that due to the time of day measurements were taken at, the velocities are likely near to the maximum daily velocity (and, later on, thus also the maximum potential for debris transport).

We have now put information on the variation of those values in the results section, in the form of a standard deviation calculated for the measurement sets. We also noted that measured flow is likely near the diurnal maximum as suggested.

L139: I can understand the logic behind the methods used to delineate stream channels and cliffs, but I do think there needs to be some evaluation of the method and/or consideration (e.g., at the very start of the Discussion section) of how the results compare to what is actually seen on the glacier surface – from field observations or from the orthophoto. Can you give a rough idea of how many are water-filled vs. abandoned?

As mentioned in the response to Reviewer #1's major comments, we plan to update the manuscript with a validation of the automated ice cliff mapping work by quantitative comparison to manually mapped ice cliff extents. Unfortunately it is quite difficult to evaluate the stream mapping work as suggested—while the orthophoto is quite good at 25 cm resolution it still may not capture small streams or channels well enough for positive identification making the delineation of water-filled vs. abandoned channels via that dataset unfeasible. As mentioned in the response to a previous comment on L58, we do have some (more qualitative) observations on the ground of supraglacial streams, which we will leverage in the discussion section in comparison with the mapping results. However, these observations are not from the same year as the orthophoto/DEM, so comparisons can't be made on specific supraglacial stream channels/networks.

L142-146: Are these statistics for full channel lengths or the 2 m sections described in L85? A reminder here of how these were calculated would allow the reader to correctly understand the data and interpretations.

These are for 2m sections. I've added to this sentence a reminder/reference to how these are calculated.

L149-150: The final clause of this sentence in an interpretation and should really be in the Discussion section.

Reviewer #1 also pointed this out; we modified and moved this language into the Discussion section.

L161: Is this stream location also confirmed by the field observations? Does it contain meltwater or has it been abandoned (i.e., active or historical undercutting)?

This location was not visited in the field, and in any case the photogrammetry was offset by 2+ years from the field observations. While we are unable to confirm whether it actively contains meltwater or has been abandoned, it is highly likely to be active due to two reasons. First is the sharp incisional morphology seen in the orthophoto/DEM, both in the undercut of the ice cliff as well as in the steep walls surrounding the predicted channel. Second is in the availability of ice cliff area draining into that channel providing meltwater to fill it.

L168 and throughout (incl. figures and supplements): The journal recommends using SI units, though perhaps there is good reason for using cm?

We can change to meters if needed. We have seen that the journal recommends SI base units (m) and accepts SI accepted units (cm), and have used cm in places to avoid small decimal numbers where features are on the scale of 1-10s cm instead of m's.

L190-202: These are interpretations and should really come in the Discussion section.

Line 200-202 and the reference to oxbow lake formation are indeed interpretations and these will accordingly be moved to the discussion section. However the rest of this text are descriptions of landscape changes observed in situ and thus will be retained in place.

L210: It could be interesting to present and discuss these statistics in two categories: 1) cliffs where the stream is immediately adjacent and actively influencing the cliff; 2) cliffs where the stream is farther away and has historically influenced the cliff. This could allow for an analysis over time of how the ice cliff profile changes as a stream moves farther away from it, though that's probably beyond the scope of this paper!

That is very interesting to think about! We have also considered this idea—that the evolution of a cliff through time will be very different for one affected by streams vs. one affected only by radiation. This is beyond the scope of the paper, but we will develop this in the discussion further as an idea for future directions.

L261: Why an increased debris thickness? General debris-covered glacier trends? If so, give references.

In this sentence we are simply describing, theoretically, what impact increasing the debris thickness would have on this system. I'm not saying there is an increased debris thickness, which would also need a spatial and/or temporal qualifier. Yes, debris thickness can be increased from moving down glacier, looking at a more heavily debris covered glacier, or perhaps moving forward in time at the same location on a debris-covered glacier. But in this sentence we are merely describing how that would impact the system, and thus I am not sure that references are necessary or even what references would be appropriate.

L269: Reference Benn et al. (2001, 10.3189/172756501781831729) for ice cliff formation from collapse of englacial conduits – and at L35 in the Introduction.

We will include this reference at both locations.

L274: Was there a difference according to ice cliff size for the order of importance of these influences?

As shown in Kneib et al. (2023)'s Supplementary Figure S10, there is no significant difference in size distribution between ice cliffs with different influences, although crevasse-originated cliffs tend to have fewer cliffs of small size.

L283: Where are cliffs more common and is there a variation in size? What could be inferred about ice cliff longevity on the basis of this spatial variation in supraglacial hydrological features?

We have not performed such an analysis in this study. An analysis of cliff size may be complicated by the fact that many of our ice cliff shapes are compounds of multiple ice cliffs (e.g. combined where separate cliffs are bordered by an ice fin). As mentioned in a response to an earlier comment, we will open up a discussion about the clear difference in supraglacial stream and ice cliff density in the active part of the glacier vs. the stagnant lower most part of the glacier. In addition, we can determine the distribution of ice cliff sizes as a function of distance from/adjacency to supraglacial streams, which can be added to Figure 4.

---

## Author Response (AR2)

Editorial comments:

L55: Move definition of NPS to here (first usage)

"NPS" was removed from here and several other locations as it is not frequently used in the text and not needed to distinguish the photogrammetry DEM presented.

L67: Add degrees to coordinates

Changed

L72: Better: 'typically tens of cm (but <50cm) thick'?

Made this suggested change.

L95: 'existence-slopes': add spaces and change hyphen to em-dash

Changed

L118: Change hyphen in 4 - 8 to en-dash and remove spaces.

Changed.

L127: vise > vice

Changed.

L220: inferring > 'implying', or 'for which we infer'

Changed to implying.

L251: In-situ > In situ

Changed.

Fig 1b: The stream gauge symbols aren't legible in panel B

Fig 1 has been re-rendered with larger stream gauge symbols in panel B.

Fig 3: State in caption what black point and arrow represent in Inset (location of a-d?)

Indicated now in caption.

Fig 6: Change superscript zeroes to degree symbol throughout figure

Changed "$^{0}$" to "$^{\circ}$"

Marin Kneib Comments:

Dear authors,
Thank you for the revision of your manuscript and your answer to my comments. I find your article to be of high quality and scientific relevance. I still had a few minor points (see below) that I feel should be addressed before publication. Line numbers correspond to the document with tracked changes.
Best regards,
Marin

L32: 'exposed ice in ice cliffs' sounds a bit weird – 'exposed ice at the surface of ice cliffs' would be a better term.

> Thank you for the suggestion. Changed to "Ice exposed at the surface of ice cliffs."

L38: No dash between 'Changri' and 'Nup', just a space.

> Fixed

L34-39: Nice inclusion, but as it is it's all a bit jumbled together with no logical order. It would make sense to distinguish modeling from purely observational studies.

> True, the writing needed some cleaning up here. I reworked these lines into the following:

> "Ice exposed at the surface of ice cliffs contributes significantly to melt on debris covered glacier surfaces \citep{sakai_distribution_2002,buri_supraglacial_2021,anderson_debris_2021}. \cite{miles_controls_2022} determined that melt rates on ice cliffs are consistently 2-3$\times$ melt rates on clean glacier ice under similar conditions. As a result, ice cliffs which cover typically $\sim$10\% of the area of debris-covered glacier tongues contribute typically $\sim$20-25\% of the melt rates over that same area, for glaciers in Alaska (\cite{anderson_debris_2021}: 12\% coverage, 26\% melt rates on Kennicott Glacier) and Nepal (\cite{brun_ice_2018}, 7-8\% area coverage, 24$\pm$5\% melt rates on Changri Nup Glacier). Ice cliff melt is also significant on glacier and catchment-wide scales; \cite{buri_supraglacial_2021} found in a modeling study that ice cliffs account for 17$\pm$4\% of total glacier melt across the Langtang Glacier catchment."

> Which makes a distinction between the scales at which each study is assessing the importance of ice cliffs. I also added a paragraph break here to make for easier reading.

L46: Well I guess then it cannot be attributed to ice dynamics only but rather to a mixture of both contributions.

I agree. In this sentence, however, I'm referring to the conclusions drawn by the authors of that study. I have clarified this in the text.

Fixed.

L68: there seems to be a missing connector between 'erode' and 'transport'
Yes – inserted "and."

L88: it is not clear here if these melt rates are for the whole glacier or just for the debris-covered part.
Just for the debris covered part (lowermost 8.5 km by length). I've rephrased the sentence based on the confusion: "Anderson et al. (2021a) estimated that 20% of Kennicott Glacier's surface is debris-covered; on the lowermost 8.5 km by length of Kennicott Glacier they determined that 26% of melt was attributed to ice cliffs covering 12% of the surface in summer 2011."

L119 & 122: if 'a threshold minimum steepest slope value was tested...' (l118), then it is more a mapping 'calibration' than 'validation'. As it is, it would be appropriate to split the manual outlines between a calibration and validation dataset to account for potential biases/overfitting issues, and to adequately quantify uncertainties of the mapping.
I do not think that there is a risk of overfitting due to the fact that the only parameter which is varied is the threshold minimum steepest slope value. Given that ice cliffs are by definition steep sections of glacier surface it makes sense that we are starting from a position of over-prediction when selecting slope angles > 30 degrees. All ice cliffs are steeper than 30 degrees (Figure 3f, "True Positive Loss"/"False Negative"), but not all pixels of the glacier surface greater than 30 degrees are ice cliff (but the probability that they are increases for higher slope angles). You make a good point that "calibration" may be more accurate than "validation." However I contend that the results of this calibration give us a good handle on mapping uncertainties through the quantification of False Positive and False Negative identifications.

L123-124: What metric did you use for the calibration?
See response to L202-207 comment. In addition to False Negative-False Positive balance and True Positive maximum, we are now quantifying F1 Score as suggested.

L152: 'himalayan' can be removed as ice sails can be found elsewhere. If needed, you can specify 'ice sails in the Karakorum' outside of the parenthesis. Note that the Himalayas do not encompass the Karakorum.

Removed "Himalayan"

L178-181: Thanks for your efforts in trying to quantify the uncertainties in the different angles. While 2 cliffs is better than none, it would strengthen the analysis if you could conduct this exercise for ~10 cliffs to be more representative.

We argue that extending this exercise as suggested is not necessary. Our results suggest that the variability between sampled cliffs is in general greater than the variability in results from sampling different locations across individual cliffs (compare Figure 6d-e to Figure S2d). Furthermore, our sampling of a large number of ice cliff-ramp systems is sure to capture much variability in terms of stream maturity, slope aspect, debris remobilization, and other possible confounding factors.

L202-207: it is common for this type of exercise to use a metric that combines TP, FP & FN (as there will in any case be an overwhelming number of TN) such as the F1- score. It would be useful to indicate this score here, as it will enable a better comparison with other methods.

Thank you for this comment! I was not familiar with the F1 score, and I was labelling what is in fact the "False Negative" with my own name the "True Positive Loss." I've re labelled that in the figure and through the text. I've calculated the F1 score (scaled to area), and it is not surprising that it agrees very closely with the results of my analysis, since my analysis was informed using all the inputs used to calculate an F1 score—TP, FP, and FN (previously referred to as "True Positive Loss" in previous drafts of this manuscript). The maximum F1 of 0.96 occurs with a minimum steepest slope angle of 51.7 degrees, which is in agreement with the maximum true positive value (93%) and 0.1 degrees difference from the value of 51.8 degrees that I elected based on matching "False Positive" to "False Negative" (previously "True Positive Loss"). This comment helps me to gain more confidence in our analysis while providing more standardized language to use. We now include the F1 score in the methods, results, and in an updated Figure 3.

Figure S3: Just to be sure, (c) shows the mean aspect of individual cliffs? This needs to be specified somewhere. I actually find it interesting to not see any obvious aspect-bias to north-facing cliffs in this figure. This could be an argument in favour of L446-447, which at the moment is only based on one particular observation (so quite weak argument) – could you perhaps look at the aspect distribution of stream-influenced and non stream-influenced cliffs to check this?

Yes, this is true for the other values as well. For clarity we added "Ice cliff metrics represent median values for ice cliff shapes, not raw pixel values."

This is an interesting idea to investigate, but goes a bit beyond the effort of this manuscript. The lack of aspect bias could also be related in large part to the crevasse-originated ice cliffs (Figure 13c), which would explain the ENE – WSW symmetry (e.g. a crevasse field oriented NNW – SSE). Again, this larger question of controls on ice cliff slope aspect goes a bit beyond the scope of this paper, so we will keep this in the supplementary material.

L227-235: this is a very nice addition!

Thank you!

Figure 10: dot missing in the caption before (b-c)

Fixed.

L388: link missing – maybe 'Such abandoned stream channel...'? L419: dot missing

Reworked and fixed.